# AutoKaggle: A Multi-Agent Framework for Autonomous Data Science Competitions

## Abstract

Data science tasks involving tabular data present complex challenges that require sophisticated problem-solving approaches. We propose AutoKaggle, a powerful and user-centric framework that assists data scientists in completing daily data pipelines through a collaborative multi-agent system. AutoKaggle implements an iterative development process that combines code execution, debugging, and comprehensive unit testing to ensure code correctness and logic consistency. The framework offers highly customizable workflows, allowing users to intervene at each phase, thus integrating automated intelligence with human expertise. Our universal data science toolkit, comprising validated functions for data cleaning, feature engineering, and modeling, forms the foundation of this solution, enhancing productivity by streamlining common tasks. We selected 8 Kaggle competitions to simulate data processing workflows in real-world application scenarios. Evaluation results demonstrate that AutoKaggle achieves a validation submission rate of 0.85 and a comprehensive score of 0.82 in typical data science pipelines, fully proving its effectiveness and practicality in handling complex data science tasks. [1]

## 1 Introduction

In recent years, with the rapid development of large language models (LLMs) (OpenAI, 2022; 2023), automated data science has gradually become possible. LLM-based agents have shown great potential in the data domain, as they can automatically understand, analyze, and process data (Hassan et al., 2023; Lucas, 2023; Zhang et al., 2024a), thereby promoting the democratization and widespread application of data science.

However, existing research still has significant shortcomings in addressing complex data science problems. Many studies are limited to simple, one-step data analysis tasks (Zhang et al., 2024c; Hu et al., 2024), which are far from the actual application scenarios of data science. While recent work (Jing et al., 2024) attempts to evaluate data science capabilities through more comprehensive tasks, it still focuses on relatively constrained scenarios that represent only portions of a complete data science pipeline. Other research relies on pre-built knowledge bases (Guo et al., 2024), raising the barrier to use and limiting the flexibility and adaptability of solutions. Moreover, current research focuses excessively on improving task completion rates and optimizing performance metrics, while neglecting the interpretability and transparency of intermediate decision-making steps in logically complex data science tasks. This neglect not only affects users' understanding of solutions but also diminishes their credibility and practicality in real-world applications.

To address these issues, we propose AutoKaggle, a universal multi-agent framework that provides data scientists with end-to-end processing solutions for tabular data, helping them efficiently complete daily data pipelines and enhance productivity. AutoKaggle has the following features:

***(i) Phase-based Workflow and Multi-agent Collaboration.*** AutoKaggle employs a phase-based workflow and multi-agent collaboration system. It divides the data science competition process into six key phases: background understanding, preliminary exploratory data analysis, data cleaning (DC), in-depth exploratory data analysis, feature engineering (FE), and model-building, -validation, and -prediction (MBVP). To execute these phases, five specialized agents (`Reader`, `Planner`,

---

[1]All code and data are available: `https://anonymous.4open.science/r/AutoKaggle-B8D2`.

`Developer`, `Reviewer`, and `Summarizer`) work collaboratively to execute these phases, from problem analysis to report generation.

***(ii) Iterative Debugging and Unit Testing.*** AutoKaggle ensures code quality through iterative debugging and unit testing. The `Developer` employs three main tools (code execution, debugging, and unit testing) to verify both syntactic correctness and logical consistency.

***(iii) Machine Learning Tools Library.*** AutoKaggle integrates a comprehensive machine learning tools library covering data cleaning, feature engineering, and model-building, -validation, and -prediction. The library includes expert-written code snippets and custom tools, enhancing code generation efficiency and quality. By combining predefined tools with self-generated code, AutoKaggle handles complex tasks while reducing reliance on LLMs for domain-specific knowledge.

***(iv) Comprehensive Reporting.*** AutoKaggle generates detailed reports after each phase and at the competition's conclusion, showcasing its decision-making process, key findings, actions, and reasoning. This feature makes the data processing workflows transparent, increasing user trust in AutoKaggle.

AutoKaggle provides a universal and comprehensive solution for a wide variety of data science tasks. By simply providing a task overview, it can automatically complete the entire process from development to testing, making it exceptionally easy to use. AutoKaggle is highly adaptable, allowing users to customize it according to their specific needs. Moreover, it offers clear interpretability throughout the automated data science process, enhancing users' understanding and trust in the system.

We chose competitions from the Kaggle platform to evaluate our framework. Kaggle data science competitions simulate the real challenges faced by data scientists, covering the complete process from data cleaning to model deployment. These competitions require participants to execute a series of complex and interdependent tasks. These include: data cleaning and preprocessing, exploratory data analysis, feature engineering, and modeling. Each step demands professional knowledge and meticulous planning, often necessitating multiple iterations. This complexity makes Kaggle an ideal platform for assessing the effectiveness of data science automation tools. In the 8 Kaggle data science competitions we evaluated, AutoKaggle achieved 0.85 in valid submission rate and 0.82 in comprehensive score. We summarize our contributions as follows:

- We propose AutoKaggle, a novel multi-agent framework for Kaggle data science competitions, achieving high task completion rates and competitive performance above the average human level in our evaluations.

- We introduce a phase-based workflow integrated with multi-agent collaboration, incorporating iterative debugging and unit testing, which systematically addresses the complexities of data science tasks and ensures robust, correct code generation.

- We develop a machine learning tools library and integrate it into our framework, enhancing code generation efficiency and quality for complex data science tasks.

- We implement a comprehensive reporting system that provides detailed insights into the decision-making process at each phase, making AutoKaggle both a solution provider and an educational tool for data science competitions, thereby contributing to the democratization of data science skills.

## 2 AUTOKAGGLE

### 2.1 OVERALL FRAMEWORK

In this section, we introduce AutoKaggle, a fully automated, robust, and user-friendly framework designed to produce directly submittable prediction results using only the original Kaggle data. Given the diversity of data science problems, the range of potential solutions, and the need for precise reasoning and real-time understanding of data changes, effectively handling complex data science tasks on Kaggle is challenging. Our technical design addresses two primary issues: *(i)* how to decompose and systematically manage complex data science tasks; and *(ii)* how to efficiently solve these tasks using LLMs and multi-agent collaboration.

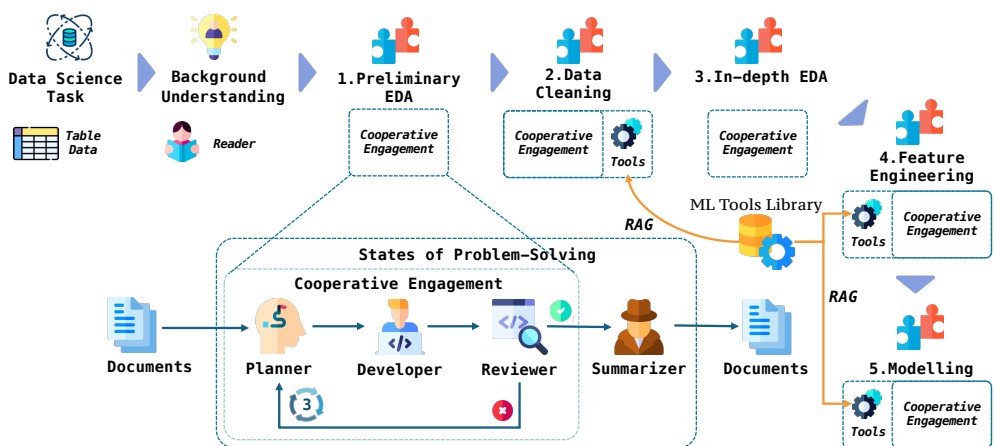

Figure 1: Overview of AutoKaggle. AutoKaggle integrates a phase-based workflow with specialized agents (`Reader`, `Planner`, `Developer`, `Reviewer`, and `Summarizer`), iterative debugging and unit testing, a comprehensive machine learning tools library, and detailed reporting.

The core concept of AutoKaggle is phase-based multi-agent reasoning. This method leverages LLMs to reason and solve tasks within a structured workflow, addressing different facets of the data science process through the collaboration of multiple agents. AutoKaggle comprises two main components: a phase-based workflow and a multi-agent system, which complement each other, as shown in Figure 1.

**Phase-based Workflow.** The data science process is divided into six key phases: understanding the background, preliminary exploratory data analysis, data cleaning, in-depth exploratory data analysis, feature engineering, and model-building, -validation, and -prediction. Data cleaning, feature engineering, and model-building, -validation, and -prediction are fundamental processes required for any data science competition. We designed two additional data analysis phases to provide essential information and insights for data cleaning and feature engineering, respectively. Given that our initial input is only an overview of a Kaggle data science competition and the raw dataset, we added a background understanding phase to analyze various aspects of the competition background, objectives, file composition, and data overview from the raw input. This structured approach ensures that all aspects of the problem are systematically and comprehensively addressed, with different phases decoupled from each other. It allows thorough unit testing at each phase to ensure correctness and prevent errors from propagating to subsequent phases.

**Multi-agent System.** The system consists of five specialized agents: `Reader`, `Planner`, `Developer`, `Reviewer`, and `Summarizer`. Each agent is designed to perform specific tasks within the workflow. They collaborate to analyze the problem, develop strategies, implement solutions, evaluate results, and generate comprehensive reports. Detailed setup and interaction processes of agents are described in Appendix D.1.

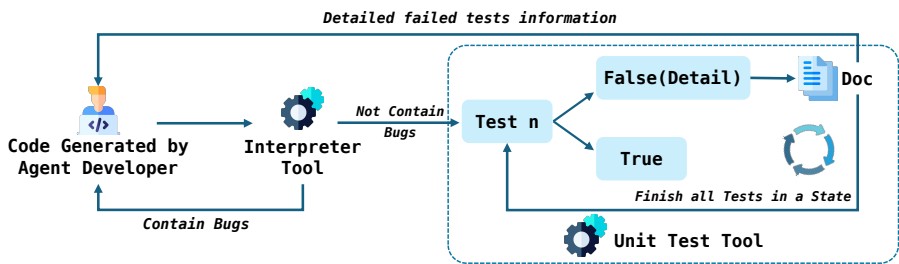

Figure 2: Iterative debugging and testing.

We summarize the pseudo-code of AutoKaggle in Algorithm 1. Let $\mathcal{C}$ represent the competition, $\mathcal{D}$ the dataset, and $\Phi = \{\phi_1, \phi_2, \ldots, \phi_6\}$ the set of all phases in the competition workflow. For each phase $\phi_i$, a specific set of agents $\mathcal{A}_{\phi_i}$ is assigned to perform various tasks. The key agents include `Planner`, `Developer`, `Reviewer`, and `Summarizer`.

## 2.2 DEVELOPMENT BASED ON ITERATIVE DEBUGGING AND TESTING

In AutoKaggle, the `Developer` adopts a development approach based on iterative error correction and testing. It ensures the robustness and correctness of generated code through iterative execution, debugging, and testing.

Figure 2 shows the overall process of iterative debugging and testing. Specifically, the `Developer` first generates code based on the current state $\mathbf{s}_t$, the plan $P_{\phi_i}$ created by the `Planner`, and the historical context $\mathcal{H}$: $C_{\phi_i} = \text{GenerateCode}(\mathbf{s}_t, P_{\phi_i}, \mathcal{H})$. $C_{\phi_i}$ is the generated code for phase $\phi_i$, and $\text{GenerateCode}(\cdot)$ represents the code generation function executed by the `Developer`. The historical context $\mathcal{H}$ includes previous phases' code, outputs, and other relevant information from other agents' activities.

After the initial code generation, it enters an iterative debugging and testing process. This process can be described by Algorithm 2.

`Developer` utilize three primary tools: code execution, code debugging, and unit testing.

*(i) Code Execution.* The Code Execution tool runs the generated code and captures any runtime errors. When an error is detected, the system restores a file to record the error messages.

*(ii) Code Debugging.* The Code Debugging tool analyzes error messages and attempts to fix the code. It utilizes error messages along with the current code and historical context to generate fixes: $C'_{\phi_i} = \text{DebugCode}(C_{\phi_i}, E_{\phi_i}, \mathcal{H})$. $C'_{\phi_i}$ is the debugged version of the code.

Following previous work (Tyen et al., 2024), we designed the debugging process into three main steps: error localization, error correction, and merging of correct and corrected code segments. We set a maximum of 5 attempts for the `Developer` to self-correct errors. Additionally, we've introduced an assistance mechanism. We record all error messages encountered during the debugging process. When the number of correction attempts reaches 3, the `Developer` evaluates the feasibility of continuing based on historical information. If past error messages are similar, it suggests that the `Developer` might lack the ability to resolve this particular error, and continuing might lead to a loop. In such cases, we allow the `Developer` to exit the correction process and regenerate the code from scratch.

*(iii) Unit Testing.* Unit testing runs predefined tests to ensure code meets requirements. For each phase $\phi_i$, a set of unit tests $T_{\phi_i}$ is defined: $T_{\phi_i} = \{t_1, t_2, \ldots, t_k\}$. The unit testing process can be represented as: $R_{\phi_i} = \text{ExecuteUnitTests}(C_{\phi_i}, T_{\phi_i})$. $R_{\phi_i}$ is the set of test results, with each result $r_j \in \{0, 1\}$ indicating whether the corresponding test passed (1) or failed (0).

In complex and accuracy-demanding tasks like Kaggle data science competitions, merely ensuring that the code runs without errors is not enough. These competitions often involve intricate data processing and sophisticated algorithms, where hidden logical errors can significantly affect the final results. Therefore, it is necessary to design meticulous unit tests that not only verify the correctness of the code but also ensure it meets the expected logical and performance standards. Otherwise, hidden errors may accumulate through successive phases, making the completion of each subsequent phase increasingly difficult. For example, unnoticed logical defects during the data cleaning phase may lead to poor feature extraction, thereby affecting the model building in subsequent phases.

To mitigate these risks, unit tests for each phase must be carefully designed to cover a wide range of scenarios, including edge cases and potential failure points. This involves not only checking the correctness of the output but also ensuring that the intermediate steps conform to the expected logic. For instance, in the data cleaning phase, unit tests should verify whether missing values are handled correctly, outliers are appropriately managed, and data transformations are accurately applied.

By implementing comprehensive unit tests, we can catch and correct errors early in the development process, preventing them from propagating to later phases. This systematic testing approach ensures

that the code at each phase is not only error-free but also functionally correct and aligned with the overall project goals.

In conclusion, the iterative debugging and testing method employed by `Developer` ensures the generation of robust, error-free, and effective code for each phase of the competition. By employing advanced error handling, iterative debugging, and comprehensive unit testing, the system can adapt to various challenges and consistently produce high-quality code outputs.

### 2.3 MACHINE LEARNING TOOLS LIBRARY

Generating machine learning code from scratch using LLMs can be challenging due to the intricacies of various tasks. These models need to encompass specialized knowledge across a range of processes, from data processing and feature engineering to model-building, -validation, and -prediction. In many cases, leveraging expert-crafted machine learning tools is more efficient than relying solely on LLM-generated code. This is because LLMs often lack domain-specific expertise, potentially leading to suboptimal or inaccurate code. Furthermore, when tasked with complex operations, the generated code may suffer from syntactical or logical errors, increasing the likelihood of failures.

Our machine learning library is categorized into three core toolsets: data cleaning, feature engineering, and model-building, -validation, and -prediction, each serving a specific role in the workflow. The data cleaning toolkit comprises seven tools, including FillMissingValues, RemoveColumns WithMissingData, DetectAndHandleOutliersZscore, DetectAndHandleOutliersIqr, RemoveDuplicates, ConvertDataTypes and FormatDatetime, all designed to ensure clean, consistent, and reliable data preparation. The feature engineering module encompasses eleven tools aimed at enhancing model performance, such as OneHotEncode, FrequencyEncode, CorrelationFeatureSelection, and ScaleFeatures, employing various techniques like correlation analysis and feature scaling to optimize data representation. The model-building, -validation, and -prediction category provides TrainAndValidationAndSelectTheBestModel to support the full model development lifecycle, including model selection, training, evaluation, prediction, ensemble integration, and hyperparameter optimization, facilitating robust model deployment and effective performance. Each tool comes with comprehensive explanations, input/output specifications, anomaly detection, and error handling guidance.

This comprehensive library is crucial for efficient multi-agent collaboration in tackling complex Kaggle competitions. Each tool provides standardized, reliable functionality, enabling AutoKaggle to seamlessly share and process data, enhance feature quality, and optimize model performance, ultimately improving overall workflow efficiency and ensuring coordinated, high-quality solutions in a competitive environment. Moreover, our machine learning library reduces the burden on AutoKaggle in detailed programming tasks, enabling them to focus more on higher-level task planning and code design. This shift of focus allows AutoKaggle to navigate complex tasks more effectively, ultimately improving their overall performance. More details of our machine learning tools can be found in Appendix D.3.

## 3 EXPERIMENTS

### 3.1 EXPERIMENTAL SETUP

**Task Selection.** We select eight Kaggle competitions that predominantly use tabular datasets, focusing on classification and regression tasks. These competitions are categorized into two types: *classic Kaggle* and *Recent Kaggle*. Classic Kaggle competitions are those that begin before October 2023 with at least 500 participants, whereas Recent Kaggle competitions begin in 2024 or later. As our analysis relies on GPT-4o, which is trained on data available until October 2023, it possibly includes information about Classic Kaggle competitions, thereby posing a risk of data leakage. To evaluate the generalization capabilities of AutoKaggle, we therefore focus on competitions initiated after 2024. Additionally, we classify these competitions into three difficulty levels: easy, medium, and hard. For each dataset, we access the corresponding competition's homepage on Kaggle, extract content from the overview and data description sections, and compile this information into a file named overview.txt. This file, along with the original competition data files, forms the primary input for AutoKaggle. More details of our datasets can be found in Appendix C.

Notably, we do not incorporate the nine tabular datasets from Mle-Bench (Hong et al., 2024) due to their substantial size, which would significantly increase computational runtime. Resource limitations prevent us from adhering to Mle-Bench's experimental setup, which specifies a 24-hour participation window per agent and a 9-hour code execution timeout.

Table 1: Made submission, valid submission and comprehensive score on 8 Kaggle tasks. Each experiment is repeated with 5 trials. The best performances on individual tasks are underlined, and the best performances across all tasks are bolded.

| Metric | Setting / Task | Classic | | | | Recent | | | | |
|---|---|---|---|---|---|---|---|---|---|---|
| | | Task 1 | Task 2 | Task 3 | Task 4 | Task 5 | Task 6 | Task 7 | Task 8 | Avg. |
| Made Submission | AutoKaggle gpt-4o | 1 | 0.80 | 0.80 | 1 | 0.80 | 0.80 | 0.80 | 0.80 | **0.85** |
| | AutoKaggle o1-mini | 1 | 0.60 | 0.60 | 1 | 0.60 | 0.80 | 0.60 | 0.60 | 0.73 |
| | AIDE gpt-4o | 1 | 0.40 | 0.20 | 0.60 | 1 | 0.80 | 0.80 | 0 | 0.60 |
| Valid Submission | AutoKaggle gpt-4o | 1 | 0.80 | 0.80 | 1 | 0.80 | 0.60 | 0.80 | 0.80 | **0.83** |
| | AutoKaggle o1-mini | 1 | 0.60 | 0.60 | 1 | 0.60 | 0.60 | 0.60 | 0.60 | 0.70 |
| | AIDE gpt-4o | 1 | 0.40 | 0.20 | 0.40 | 1 | 0.80 | 0.80 | 0 | 0.58 |
| Comprehensive Score | AutoKaggle gpt-4o | 0.888 | 0.786 | 0.831 | 0.862 | 0.810 | 0.728 | 0.848 | 0.812 | **0.821** |
| | AutoKaggle o1-mini | 0.879 | 0.680 | 0.729 | 0.863 | 0.709 | 0.735 | 0.742 | 0.735 | 0.759 |
| | AIDE gpt-4o | 0.872 | 0.597 | 0.542 | 0.561 | 0.918 | 0.793 | 0.848 | 0 | 0.641 |

**Evaluation metric.** We evaluate the capability of the AutoKaggle from four perspectives: Made Submission, Valid Submission, Average Normalized Performance Score and Comprehensive Score. The first two metrics refer to Mle-bench and are primarily used to assess the ability to generate a submission.csv file. The last two metrics come from Data Interpreter (Chan et al., 2024), we made modifications to adapt them to the evaluation of our framework.

*(i) Made Submission (MS).* Made Submission refers to the percentage of times a submission.csv file is generated.

*(ii) Valid Submission (VS).* Valid Submission indicates the percentage of those submission.csv files that are valid—meaning they can be successfully submitted to the Kaggle website, produce results without errors, and have no issues related to data scale or category mismatches.

*(iii) Comprehensive Score (CS).* In the evaluations, performance metrics are divided into two categories: bounded metrics, which range from 0 to 1 where higher values indicate better performance, and unbounded metrics, where lower values denote superior performance. To normalize these different types of metrics, we utilize the normalized performance score (NPS), defined as follows:

$$\text{NPS} = \begin{cases} \frac{1}{1+s}, & \text{if } s \text{ is smaller the better} \\ s, & \text{otherwise.} \end{cases} \quad (1)$$

For multiple trials of a task, we calculate the Average Normalized Performance Score (ANPS) as the average of the successful attempts:

$$\text{ANPS} = \frac{1}{T_s} \sum_{t=1}^{T_s} \text{NPS}_t \quad (2)$$

Table 2: Ablation study on machine learning tools. Evaluated with completion rate and comprehensive score. Best performance are underlined.

| | | Task 1 | Task 2 | Task 3 | Task 5 | Avg. |
|---|---|---|---|---|---|---|
| VS | No Tools | 0.80 | 0.60 | 0.50 | 0.40 | 0.58 |
| | DC Tools | 0.80 | 0.70 | 1.00 | 1.00 | **0.88** |
| | DC & FE Tools | 0.80 | 0.60 | 0.60 | 0.60 | 0.65 |
| | All Tools | 1.00 | 0.80 | 0.80 | 0.80 | 0.85 |
| CS | No Tools | 0.781 | 0.697 | 0.666 | 0.602 | 0.687 |
| | DC Tools | 0.781 | 0.721 | 0.928 | 0.909 | **0.835** |
| | DC & FE Tools | 0.787 | 0.684 | 0.735 | 0.713 | 0.730 |
| | All Tools | 0.888 | 0.786 | 0.831 | 0.810 | 0.829 |

where $T_s$ represents the total number of successful attempts for a task, and $\text{NPS}_t$ is the NPS value for the $t$-th attempt.

To comprehensively evaluate both the pass rate and the average performance, we define the Comprehensive Score (CS) as the average of VS and ANPS:

$$\text{CS} = 0.5 \times \text{VS} + 0.5 \times \text{ANPS} \tag{3}$$

**Experiment Details.** We evaluated AutoKaggle's performance based on both GPT-4o and o1-mini models. Notably, different models were assigned to specific agents based on their functional requirements. The `Reader`, `Reviewer`, and `Summarizer`, which perform tasks requiring minimal logical reasoning and coding capabilities, were implemented using the GPT-4o-mini model. The `Planner`, responsible for task decomposition and planning that demands sophisticated logical reasoning, operates on either the GPT-4o or o1-mini model. Although the `Developer`'s tasks traditionally necessitate advanced logical reasoning and coding skills, the `Planner`'s effective task decomposition methodology has moderated these requirements, therefore it is based on GPT-4o model.

In our experiments, Each task undergoes five trials, with each phase in the workflow allowing for a maximum of three iterations. During an iteration, the `Developer` may debug the code up to five times. If unsuccessful, they proceed with the same phase, deriving insights and adjusting strategies based on previous attempts. Failure to resolve issues after three iterations is considered a definitive failure.

**Baseline.** We employ AIDE (Schmidt et al., 2024) as our baseline, which is the best-performing framework in Mle-bench evaluation results. We use AIDE's default settings, only modifying `agent.base.model` to the GPT-4o model.

## 3.2 MAIN RESULTS

The comprehensive performance of AutoKaggle across 8 Kaggle data science competitions is presented in Table 1. In order to facilitate understanding, we uniformly name the eight tasks as task 1-8. The real task names and detailed dataset information are available in Appendix C.

**Average NPS for Different Settings/Tasks**

■ AutoKaggle-gpt-4o ■ AutoKaggle-o1-mini ■ AIDE-gpt-4o

Figure 3: Average normalized performance score for different settings/tasks.

**Made submission and Valid submission.** We first evaluated the success rate of valid submission.csv file generation across different experimental configurations. The AutoKaggle framework, implemented with GPT-4o, demonstrated superior performance with an average valid submission

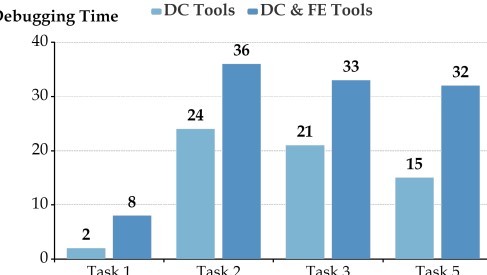 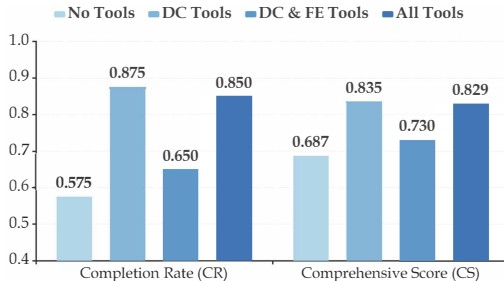

Figure 4: **Left.** Debugging time and **Right.** Average performance in competitions.

rate of 83% across all 8 Kaggle tasks, surpassing the AIDE framework by 28%. These results underscore the robustness of our framework in executing comprehensive data science workflows. While the AIDE framework successfully processed Tasks 1-7, which involved single-variable classification or regression on tabular data, it failed to generate valid submissions for Task 8, a multi-variable classification problem. This differential performance demonstrates our framework's versatility in handling diverse tabular data tasks.

Another interesting observation is that within the AutoKaggle framework, the GPT-4o model achieved better results than the o1-mini model, despite the latter's purported superior reasoning capabilities. This performance difference emerged solely from varying the model used in the Planner component. We hypothesize that this counterintuitive result stems from o1-mini's tendency toward excessive planning complexity, which proves disadvantageous in our streamlined, phase-based workflow architecture. This same consideration influenced our decision to maintain GPT-4o as the `Developer`'s base model, as our experiments indicated that an o1-mini-based `Developer` would significantly increase code verbosity, expanding 100-line solutions to approximately 500 lines through the introduction of superfluous components such as logging systems.

**Comprehensive Score.** Subsequently, we compared the overall performance of different settings across 8 Kaggle tasks. AutoKaggle with GPT-4o achieved the highest comprehensive score in 5 tasks and demonstrated the best overall performance. Figure 3 illustrates the comparison of different settings based on the average normalized performance score metric, where AutoKaggle with o1-mini achieved the highest overall score. This indicates that although the o1-mini-based `Planner` generated overly complex plans that increased development difficulty, successfully executing these plans according to specifications led to superior performance outcomes.

## 3.3 ABLATION STUDY

Apart from the modules involved in the ablation study, all other experimental settings are identical to those in the formal experiment.

**Study on Machine Learning Tools.** To evaluate the effectiveness of the machine learning tools module and the impact of tools across different phases on the results, we conduct ablation experiments. We begin without any tools and progressively add them at each phase until all machine learning tools are implemented. The results are presented in Table 2. Notably, the completion rate increases by 30% with the use of data cleaning phase tools, and by 27.5% when all tools are utilized, compared to the scenario with no tools. However, the completion rate exhibits a decline during the feature engineering phase, particularly in the house prices and academic success competitions. This decline can be attributed to the relatively large number of features involved, alongside the complexity and high encapsulation of the tools used in this phase, which necessitate the addition and removal of features, thereby complicating their usage. Furthermore, this complexity poses challenges for `Developers` in debugging erroneous code. As illustrated in Figure 4 (a), the frequency of debugging instances is greater when employing tools from the feature engineering phase.

Figure 4 (b) provides a clearer comparison, demonstrating that while the best normalized performance scores across four scenarios are similar, the completion rate significantly increases with the use of the tool. This suggests that although the machine learning tool library we develop does not substantially elevate the solution's upper limit, it functions as a more stable tool that enhances AutoKaggle's completion rate. This outcome aligns with expectations, as the machine learning tool

library is a redevelopment based on widely used libraries such as pandas and scikit-learn. It does not introduce new functionalities but instead combines and re-packages existing ones, incorporating error handling and manual testing to ensure compatibility with our framework.

**Study on Unit Tests.** To evaluate the effectiveness of the unit tests module, we conduct ablation experiments. The results are presented in Table 3. In the absence of unit tests, the completion rate significantly decreases, making it nearly impossible to complete the tasks. This emphasizes that for tasks like data science, which demand high levels of precision and logic, it is not enough for each phase of the code to merely execute without errors. Comprehensive unit testing is required to ensure that the code is logical and achieves the objectives of each phase.

**Study on Debugging Times.** We conduct ablation experiments to investigate the impact of the number of allowed debugging times on the results. The experimental setup permits five code debugging attempts within each phase, with each phase being executable up to three times. Consequently, we analyze scenarios with allowable corrections set at 0, 5, and 10. The results are shown in Figure 5. It can be observed that when AutoKaggle is required to pass without any errors, there is only one successful record on the Titanic task. Allowing five debugging attempts significantly improves the completion rate, and further increases in allowable debugging attempts lead to rises in all metrics. This demonstrates the efficacy of our code debugging module. However, the performance when the number of allowable debugging attempts is set to 10 and 15, suggesting that the agent's self-correction abilities are limited. There are complex errors that it cannot resolve independently, and further increasing the number of allowable debugging attempts does not address these errors.See more details in Section B.

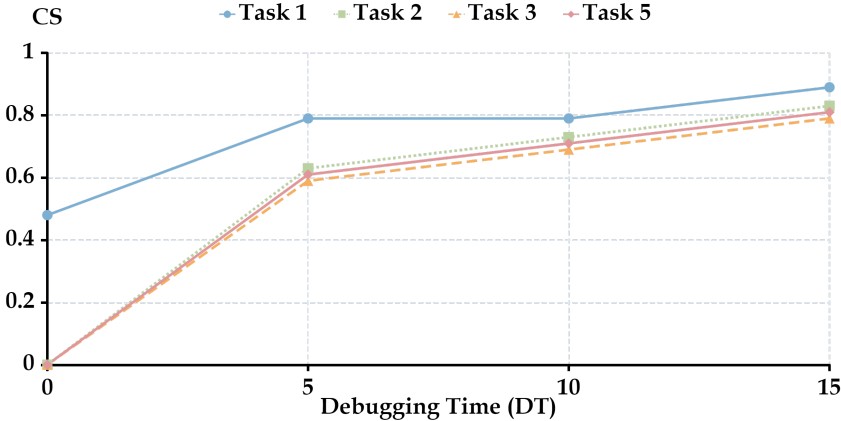

Figure 5: Comprehensive Score across different debugging times.

**Study on Competition Date.** To further evaluate the generalization capabilities of our AutoKaggle framework, we conducted an analysis stratified by competition dates. Tasks 1-4 corresponded to competitions potentially included in the training data of models such as GPT-4o and O1-mini, while tasks 5-8 were derived from competitions launched in the current year. This temporal stratification enabled us to assess the framework's performance on out-of-distribution tasks. For classic Kaggle tasks, AutoKaggle with GPT-4o achieved a valid submission rate of 0.90 and a comprehensive score of 0.842. On recent tasks, these metrics were 0.75 and 0.800 respectively, demonstrating only marginal performance degradation. These results indicate that our task decoupling approach and

Table 3: Ablation study on unit tests. Better performance are underlined.

|     |                | Task 1 | Task 2 | Task 3 | Task 5 | Avg.  |
|-----|----------------|--------|--------|--------|--------|-------|
| CR  | w/o Unit Tests | 0.20   | 0      | 0.20   | 0      | 0.10  |
|     | w/ Unit Tests  | 1.00   | 0.80   | 0.80   | 0.80   | 0.85  |
| CS  | w/o Unit Tests | 0.478  | 0      | 0.482  | 0      | 0.240 |
|     | w/ Unit Tests  | 0.888  | 0.831  | 0.786  | 0.810  | 0.829 |

predefined execution pathways enable effective handling of novel competitions, even in scenarios where the underlying model lacks prior exposure to the domain.

## 4 RELATED WORK

A concise framework of agents consists of brain, perception, and action modules (Xi et al., 2023). The perception module processes external information, the brain plans based on that information, and the action module executes these plans (Xi et al., 2023; Zhou et al., 2023). LLMs, acting as brain modules, exhibit impressive zero-shot abilities and are applied in fields like data science and music composition (Brown et al., 2020; Hong et al., 2024; Deng et al., 2024). While the chain-of-thought method enhances reasoning (Wei et al., 2023), it still faces challenges related to hallucinations and unfaithfulness (Turpin et al., 2023), potentially due to internal representations (Yao et al., 2023). The ReAct paradigm addresses this by integrating thoughts and actions, refining outputs through interaction with external environments (Yao et al., 2023; Madaan et al., 2023; Shinn et al., 2023; Zhou et al., 2024).

While an individual agent can achieve basic natural language processing (NLP) tasks, real-world tasks have higher complexities. In human societies, people chunk complex tasks into simple subtasks that different people can easily handle. Inspired by this division of labor principle, multi-agent systems enhance performance (Talebirad & Nadiri, 2023) using cooperative interactions (Xi et al., 2023; Li et al., 2023) to achieve shared goals. Another interaction method is adversarial interactions (Lewis et al., 2017), where several agents compete with each other for better results, or one agent critiques and reviews the generation of another agent (Gou et al., 2024).

Previous studies have similarly focused on addressing problems in the data science domain, but many of these approaches suffer from limited scalability due to heavy reliance on pre-constructed expert knowledge bases (Guo et al., 2024) or the need for historical data as experience pools (Zhang et al., 2024a). Recently, the AIDE (Schmidt et al., 2024) framework demonstrated strong performance in Mle-Bench(Chan et al., 2024). However, its solutions face challenges such as insufficient process transparency and significant deviations from human logical reasoning, limiting their interpretability and generalizability.

In comparison, AutoKaggle adopt hierarchical systems (Hong et al., 2024; Zhang et al., 2024b; Chi et al., 2024) to complete tasks such as task understanding, feature engineering, and model building. In each hierarchy, separately design two agents for the code planning and code generation respectively (Hong et al., 2024), and use unit tests (Zhang et al., 2024b) to verify the quality of code generation. Beyond self-debugging by autonomous multi-agents, human-in-the-loop (Hong et al., 2024; Zhang et al., 2024b) mechanisms also provide oversight and corrections to LLM outputs, reducing hallucinations in each hierarchy.

In summary, multi-agent systems and LLM-based agents have demonstrated significant potential across domains such as NLP and data science. While individual agents excel in basic tasks, integrating multiple agents is crucial for tackling complex real-world challenges. By combining task-specific agents with human-in-the-loop mechanisms and unit testing, these systems improve code quality and address issues like hallucinations. Our framework, AutoKaggle, advances these efforts by integrating LLM-based reasoning with multi-agent collaboration, ensuring adaptability, correctness, and user control in data science competitions.

## 5 CONCLUSION

In this paper, we introduce AutoKaggle, a robust framework designed to leverage phase-based workflows and multi-agent collaboration for solving complex Kaggle data science competitions. AutoKaggle employs an iterative development process, incorporating thorough code debugging, unit testing, and a specialized machine learning tools library to address the intricate requirements of data science tasks. Our framework enhances reliability and automation in managing sophisticated data workflows, while maintaining user control through customizable processes. Extensive evaluations across various Kaggle competitions demonstrate AutoKaggle's effectiveness, marking a significant advancement in AI-assisted data science problem-solving and expanding the capabilities of LLM-based systems in tackling real-world challenges.

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

# A ALGORITHM

---

**Algorithm 1:** AutoKaggle Workflow

---

**Input** : Competition $\mathcal{C}$, Dataset $\mathcal{D}$
**Output:** Solution $\mathcal{S}$, Comprehensive report $\mathcal{R}$

1 Initialize state $\mathbf{s}_0$ with first phase $\phi_1$: "Understand Background";
2 $t \leftarrow 0$;
3 $\Phi \leftarrow \{\phi_1, \phi_2, ..., \phi_6\}$ ;                    /* Set of all phases */
4 Define $\mathcal{A}_\phi$ for each $\phi \in \Phi$ ;                    /* Agents for each phase */
5 **do**
6     $\mathbf{s}_t \leftarrow \text{GetCurrentState}()$;
7     $\phi_{\text{current}} \leftarrow \text{GetCurrentPhase}(\Phi)$;
8     $\mathcal{A}_t \leftarrow \mathcal{A}_{\phi_{\text{current}}}$;
9     **for** $a \in \mathcal{A}_t$ **do**
10        **if** *a is Planner* **then**
11           $r_a \leftarrow a.\text{execute}(\mathbf{s}_t)$;
12           $\mathbf{s}_t \leftarrow \text{UpdateState}(\mathbf{s}_t, r_a)$;
13           **if** *UserInteractionEnabled*() **then**
14              $\mathbf{s}_t \leftarrow \text{UserReview}(\mathbf{s}_t)$ ;                    /* User Review plan */
15        **else if** *a is Developer* **then**
16           $r_a \leftarrow a.\text{execute}(\mathbf{s}_t)$;
17           $\mathbf{s}_t \leftarrow \text{UpdateState}(\mathbf{s}_t, r_a)$;
18           **if** *NoErrors*($r_a$) **then**
19              $T \leftarrow \text{ExecuteUnitTests}(\phi_{\text{current}})$;
20              **if** $\neg$*PassTests*($T$) **then**
21                 $\mathbf{s}_t \leftarrow \text{Debug}(\mathbf{s}_t)$;
22        **else**
23           $r_a \leftarrow a.\text{execute}(\mathbf{s}_t)$;
24           $\mathbf{s}_t \leftarrow \text{UpdateState}(\mathbf{s}_t, r_a)$;
25     **if** *AllAgentsCompleted*($\mathcal{A}_t$) **and** *PassTests*($T$) **then**
26        $\phi_{\text{current}} \leftarrow \text{NextPhase}(\Phi)$;
27     $t \leftarrow t + 1$;
28 **while** $\exists \phi \in \Phi$ : *not completed*($\phi$);
29 $\mathcal{S} \leftarrow \text{ExtractSolution}(\mathbf{s}_t)$;
30 $\mathcal{R} \leftarrow \text{GenerateReport}(\mathbf{s}_t)$;

---

---

**Algorithm 2:** Development based on Iterative Debugging and Testing

---

**Input** : Initial code $C_{\phi_i}$, Current state $\mathbf{s}_t$, Plan $P_{\phi_i}$, Historical context $\mathcal{H}$, Maximum tries $max\_tries$, Error threshold $threshold$

**Output:** Debugged and tested code $C'_{\phi_i}$, Execution flag $execution\_flag$

1 $round \leftarrow 0$;
2 $error\_flag \leftarrow false$;
3 $execution\_flag \leftarrow true$;
4 $retry\_flag \leftarrow false$;
5 $error\_history \leftarrow \emptyset$;
6 **while** $round < max\_tries$ **do**
7     **if** $round = 0$ **or** $retry\_flag$ **then**
8         $C_{\phi_i} \leftarrow$ GenerateCode($\mathbf{s}_t, P_{\phi_i}, \mathcal{H}$);
9         $error\_history \leftarrow \emptyset$;
10         $retry\_flag \leftarrow false$;
11     $error\_flag, E_{\phi_i} \leftarrow$ RunCode($C_{\phi_i}$);
12     **if** $error\_flag$ **then**
13         $error\_history \leftarrow error\_history \cup \{E_{\phi_i}\}$;
14         **if** $|error\_history| >= threshold$ **then**
15             $retry\_flag \leftarrow$ EvaluateRetry($error\_history$);
16             **if** $retry\_flag$ **then**
17                 **continue**;
18         $C_{\phi_i} \leftarrow$ DebugCode($C_{\phi_i}, E_{\phi_i}, \mathcal{H}$);
19     **else**
20         $R_{\phi_i} \leftarrow$ ExecuteUnitTests($C_{\phi_i}, T_{\phi_i}$);
21         **if** $\exists r_j \in R_{\phi_i} : r_j = 0$ **then**
22             $C_{\phi_i} \leftarrow$ DebugTestFailures($C_{\phi_i}, R_{\phi_i}, \mathcal{H}$);
23         **else**
24             $execution\_flag \leftarrow true$;
25             **break**;
26     $round \leftarrow round + 1$;
27 **if** $round = max\_tries$ **then**
28     $execution\_flag \leftarrow false$;
29 **return** $C_{\phi_i}, execution\_flag$

---

## B    ERROR ANALYSIS

In each subtask phase of AutoKaggle, errors may occur, with data cleaning and feature engineering experiencing the highest error rates at 25% and 22.5%, respectively. Notably, failures during the feature engineering phase result in direct competition failures in 31.25% of cases.

In the context of the proposed AutoKaggle framework, which aims to assist data scientists in solving complex tabular data challenges through a collaborative multi-agent system, Table 4 provides an overview of the different types of errors encountered during the iterative development process. AutoKaggle's workflow includes code execution, debugging, and comprehensive unit testing, and the listed errors are indicative of the various challenges encountered while automating these stages. The most frequently observed errors are Value Errors (49 occurrences), related to mismatched input types or ranges, and Key Errors (44 occurrences), resulting from attempts to access non-existent dictionary keys. Additionally, Type Errors (25 occurrences) and Model Errors (8 occurrences) highlight operational issues due to data type mismatches or incorrect model configurations, respectively. The table also details other errors such as Timeout, FileNotFound, and Index Errors, each contributing to the debugging process. Understanding these error types is crucial for improving AutoKaggle's robustness and aligning automated workflows with human interventions, ultimately enhancing productivity in typical data science pipelines.

In addition, we provide a detailed debugging process for developers. Below, we illustrate this using a FileNotFoundError as an example of the debugging workflow:

- **Error Localization**: The developer initially encounters issues executing a Python script involving file-saving operations with libraries like Matplotlib and Pandas. The specific error, FileNotFoundError, is traced to nonexistent directories or incorrect file paths. Through an iterative analysis, the problematic sections of the code are identified, focusing on the need to properly manage directory paths and handle filenames.

- **Error Correction**: To address these issues, several modifications are suggested. First, the importance of ensuring that directories exist before performing file operations is highlighted by incorporating os.makedirs to create any missing directories. Additionally, a filename sanitization approach is recommended to prevent errors related to invalid characters in file paths. A custom sanitize_filename function is introduced to ensure filenames contain only valid characters, thereby avoiding issues caused by special symbols or whitespace.

- **Merging Correct and Corrected Code Segments**: The final step involves merging the corrected segments back into the original code to create a seamless and robust solution. The revised script includes improvements such as verifying directory existence, creating necessary directories, and applying filename sanitization to ensure compatibility across different operating systems. The corrected code is delivered with a focus on enhancing reliability, particularly in file-saving processes, making it resilient against common pitfalls like missing directories or invalid filenames.

## C  DETAILED DATASET DESCRIPTION

Here is the detailed description of our dataset. Note that we use task labels to represent the different datasets. Task 1 refers to Titanic (Cukierski, 2012), Task 2 refers to Spaceship Titanic (Addison Howard, 2022), Task 3 refers to House Prices (Anna Montoya, 2016), Task 4 refers to Monsters (Kan, 2016), Task 5 refers to Academic Success (Walter Reade, 2024d), Task 6 refers to Bank Churn (Walter Reade, 2024a), Task 7 refers to Obesity Risk (Walter Reade, 2024b), and Task 8 refers to Plate Defect (Walter Reade, 2024c).

Our framework deliberately avoids selecting competitions with excessively large datasets. The reason for this is that larger datasets significantly extend the experimental runtime, making it impractical to dedicate a machine to a single experiment for such prolonged periods.

Table 4: Error Types of AutoKaggle in the Problem-Solving Stage

| Error Type (Count) | Description |
|---|---|
| Value Error (49) | Fail to match the expected type or range of the input values |
| Key Error (44) | Attempt to access a dictionary element using a key that does not exist |
| File Error (8) | Attempt to access a file that does not exist in the specified location |
| Model Error (8) | Incorrect setup in the parameters or structure of a model, leading to operational failures |
| Type Error (25) | Mismatch between expected and actual data type, leading to operational failure |
| Timeout Error (6) | Failure to complete a process within the allocated time period |
| Index Error (3) | Attempt to access an element at an index that is outside the range of a list or array |
| Assertion Error (1) | An assertion condition in the code is not met, indicating an unmet expected constraint |
| Name Error (2) | Use of an undeclared variable that is not recognized by the system |
| Attribute Error (2) | Attempt to access an attribute or method that does not exist for an object |
| Indentation Error (1) | Incorrect indentation disrupts code structure, preventing proper parsing |

First, we intentionally avoided selecting competitions with datasets that were too large, as larger datasets can significantly extend the experimental runtime, making it impractical to use a single machine for extended experiments. Second, we adhered to real-world competition settings by generating submission files and submitting them manually for evaluation. Simply splitting the training data would result in a test set with a distribution very similar to the training data, which could inflate performance metrics—similar to the difference often seen between validation scores and real test scores. Third, our dataset clearly identifies the contest type, i.e., tabular data. Fourth, since datasets for large language modeling include publicly available Kaggle contests, we selected only those released after 2024. Our framework requires agents to independently interpret contest tasks, understand the data, and determine appropriate optimization strategies without relying on predefined guidance."

Table 5: Selected Kaggle tasks. For each task, we show its number, category, difficulty level, number of teams, train size and test size in dataset.

| Category | No. | Task Name | Task | Level | Teams | Train | Test |
|---|---|---|---|---|---|---|---|
| **Classic** | 1 | Titanic | Classification | Medium | 13994 | 891 | 418 |
| | 2 | Spaceship Titanic | Classification | Easy | 1720 | 8693 | 4277 |
| | 3 | House Prices | Regression | Medium | 4383 | 1460 | 1459 |
| | 4 | Monsters | Classification | Easy | 763 | 371 | 529 |
| **Recent** | 5 | Academic Success | Regression | Medium | 2684 | 76.5K | 51K |
| | 6 | Bank Churn | Regression | Easy | 3632 | 165K | 110K |
| | 7 | Obesity Risk | Classification | Easy | 3587 | 20.8K | 13.8K |
| | 8 | Plate Defect | Regression | Medium | 2199 | 19.2K | 12.8K |

## D  IMPLEMENTATION DETAILS

### D.1  AGENT DETAILS

#### D.1.1  AGENT BASE

The base agent is a father class of other agents (`Reader`, `Planner`, `Developer`, `Reviewer`, and `Summarizer`) in the AutoKaggle. This agent can act with various tools for tasks related to data analysis, model evaluation, and document retrieval etc.

#### D.1.2  READER

`Reader` is designed for reading documents and summarizing information. It processes overview.txt in each competition, subsequently providing a well-organized summary of the competition's background

---

**Prompt of Agent Reader / Task Prompt**

**Role:** reading documents and summarizing information
**Description:** The Reader only appears in the Understand Background phase, it reads the overview.txt file of the Kaggle competition, the sample data of both training and testing sets and summarizes it into a clearly structured competition_info.txt in markdown format.

---

**Prompt of Agent Reader / Task Prompt**

```
# CONTEXT #
{phases_in_context}
Currently, I am at phase: Background Understand.

#############
# TASK #
{task}

#############
# RESPONSE #
Let's work this out in a step by step way.

#############
# START ANALYSIS #
If you understand, please request the overview of this data science competition, and
    data preview from me.
```

```
Please conduct a comprehensive analysis of the competition, focusing on the following
    aspects:
1. Competition Overview: Understand the background and context of the topic.
2. Files: Analyze each provided file, detailing its purpose and how it should be used
    in the competition.
3. Problem Definition: Clarify the problem's definition and requirements.
4. Data Information: Gather detailed information about the data, including its
    structure and contents.
    4.1 Data type:
        4.1.1. ID type: features that are unique identifiers for each data point,
            which will NOT be used in the model training.
        4.1.2. Numerical type: features that are numerical values.
        4.1.3. Categorical type: features that are categorical values.
        4.1.4 Datetime type: features that are datetime values.
    4.2 Detailed data description
5. Target Variable: Identify the target variable that needs to be predicted or
    optimized, which is provided in the training set but not in the test set.
6. Evaluation Metrics: Determine the evaluation metrics that will be used to assess
    the submissions.
7. Submission Format: Understand the required format for the final submission.
8. Other Key Aspects: Highlight any other important aspects that could influence the
    approach to the competition.
Ensure that the analysis is thorough, with a strong emphasis on :
1. Understanding the purpose and usage of each file provided.
2. Figuring out the target variable and evaluation metrics.
3. Classification of the features.
```

### D.1.3 PLANNER

`Planner` is designed for creating task plans and roadmaps. The agent's main function is to structure and organize tasks into executable plans, primarily by leveraging available tools and previously generated reports.

**Prompt of Agent Planner / Task Prompt**

```
Role:  creating task plans and roadmaps
Description:  In the first execution, the Planner collects
the competition information, the current state, and the
user's rules to generate a new plan.  This generation
involves several rounds of interaction with a LLM to gather
task details, reorganize data into structured formats
(Markdown and JSON), and finalize a plan.
```

**Prompt of Agent Planner / Task Prompt**

```
# CONTEXT #
{phases_in_context}
Currently, I am at phase: {phase_name}.

#############
# INFORMATION #
{background_info}

{state_info}

#############
# NOTE #
## PLANNING GUIDELINES ##
1. Limit the plan to a MAXIMUM of FOUR tasks.
2. Provide clear methods and constraints for each task.
3. Focus on critical steps specific to the current phase.
4. Prioritize methods and values mentioned in USER RULES.
5. Offer detailed plans without writing actual code.
6. ONLY focus on tasks relevant to this phase, avoiding those belonging to other
     phases. For example, during the in-depth EDA phase:
    - you CAN perform detailed univariate analysis on KEY features.
    - you CAN NOT modify any feature or modify data.

## DATA OUTPUT PREFERENCES ##
1. Prioritize TEXT format (print) for statistical information.
2. Print a description before outputting statistics.
3. Generate images only if text description is inadequate.

## METHODOLOGY REQUIREMENTS ##
1. Provide highly detailed methods, especially for data cleaning.
2. Specify actions for each feature without omissions.

## RESOURCE MANAGEMENT ##
1. Consider runtime and efficiency, particularly for:
    - Data visualization
    - Large dataset handling
    - Complex algorithms
2. Limit generated images to a MAXIMUM of 10 for EDA.
3. Focus on critical visualizations with valuable insights.

## OPTIMIZATION EXAMPLE ##
When using seaborn or matplotlib for large datasets:
- Turn off unnecessary details (e.g., annot=False in heatmaps)
- Prioritize efficiency in plot generation

#############
# TASK #
{task}

#############
# RESPONSE #
Let's work this out in a step by step way.

#############
# START PLANNING #
Before you begin, please request the following documents from me, which contain
     important information that will guide your planning:
1. Report and plan from the previous phase
2. Available tools in this phase
3. Sample data for analysis
```

```
Please design plan that is clear and specific to each FEATURE for the current
     development phase: {phase_name}.
The developer will execute tasks based on your plan.
I will provide you with INFORMATION, RESOURCE CONSTRAINTS, and previous reports and
     plans.
You can use the following reasoning pattern to design the plan:
1. Break down the task into smaller steps.
2. For each step, ask yourself and answer:
    - "What is the objective of this step?"
    - "What are the essential actions to achieve the objective?"
    - "What features are involved in each action?"
    - "Which tool can be used for each action? What are the parameters of the tool?"
    - "What are the expected output of each action? What is the impact of the action
         on the data?"
    - "What are the constraints of this step?"
```

### D.1.4 DEVELOPER

`Developer` is responsible for implementing and debugging code based on the structured plans generated by the Planner. The `Developer`'s key function is to translate the high-level task roadmap into executable code, resolve any arising issues, and perform unit tests to ensure the functionality of the solution.

---

**Prompt of Agent Developer / Task Prompt**

```
Role:  write and implement code according to plan
Description:  The Developer first reviews the task plan
and the relevant competition information.  It can gathers
code from previous phases when necessary and uses LLMs
to generate new code.  The Developer also cleans up any
redundant code sections, writes functions, and ensures the
code runs correctly by debugging and performing unit tests.
It iterates through the process until the code passes all
tests.
```

---

**Prompt of Agent Developer / Task Prompt**

```
# CONTEXT #
{phases_in_context}
Currently, I am at phase: {phase_name}.

#############
# INFORMATION #
{background_info}

{state_info}

#############
# PLAN #
{plan}

#############
# TASK #
{task}

#############
# RESPONSE: BLOCK (CODE & EXPLANATION) #
TASK 1:
THOUGHT PROCESS:
[Explain your approach and reasoning]
CODE:
'''python
[code]
'''
EXPLANATION:
[Brief explanation of the code and its purpose]

TASK 2:
[Repeat the above structure for each task/subtask]

...

#############
# START CODING #
Before you begin, please request the following information from me:
1. Code from previous phases
2. All features of the data
3. Available tools

Once you have this information, provide your complete response with code and
    explanations for all tasks in a single message.
```

```
Develop an efficient solution based on the Planner's provided plan:
1. Implement specific tasks and methods outlined in the plan
2. Ensure code is clear, concise, and well-documented
3. Utilize available tools by calling them with correct parameters
4. Consider data types, project requirements, and resource constraints
5. Write code that is easily understandable by others

Remember to balance efficiency with readability and maintainability.
```

### D.1.5 REVIEWER

`Reviewer` is responsible for evaluating the performance of other agents in completing tasks and providing constructive feedback.

**Prompt of Agent Reviewer / Task Prompt**

**Role:** assess agent performance and offer feedback
**Description:** The Reviewer agent evaluates the performance of multiple agents. In each evaluation phase, it merges suggestions and scores from different agents into a unified report. It interacts with a LLM to generate detailed feedback, iterating through rounds to assess task results, merging agent responses, and producing both final scores and constructive suggestions.

**Prompt of Agent Reviewer**

```
# CONTEXT #
{phases_in_context}
Each phase involves collaboration between multiple agents. You are currently
    evaluating the performance of agents in Phase: {phase_name}.

#############
# TASK #
Your task is to assess the performance of several agents in completing Phase: {
    phase_name}.
I will provide descriptions of each agent, the tasks they performed, and the outcomes
    of those tasks.
Please assign a score from 1 to 5 for each agent, with 1 indicating very poor
    performance and 5 indicating excellent performance.
Additionally, provide specific suggestions for improving each agent's performance, if
    applicable.
If an agent's performance is satisfactory, no suggestions are necessary.

#############
# RESPONSE: JSON FORMAT #
Let's work this out in a step by step way.

#############
# START EVALUATION #
If you are ready, please request from me the role, description, input, task and
    execution result of the agent to be evaluated.
```

### D.1.6 SUMMARIZER

`Summarizer` is responsible for generating summaries, designing questions, and reorganizing both questions and answers to produce structured reports based on the competition phases.

**Prompt of Agent Summarizer / Task Prompt**

**Role:** assess agent performance and offer feedback
**Description:** The agent Summarizer works through various phases, each focusing on a specific task like choosing relevant images, designing key questions, answering phase-related questions, and organizing the responses into a structured report. Each phase involves interaction with provided inputs such as competition information, the planner's plan, and the reviewer's evaluation to synthesize the most relevant insights.

**Prompt of Agent Summarizer**

```
# TASK #
Please reorganize the answers that you have given in the previous step, and synthesize
    them into a report.

#############
# RESPONSE: MARKDOWN FORMAT #
```markdown
# REPORT
## QUESTIONS AND ANSWERS
### Question 1
What files did you process? Which files were generated? Answer with detailed file path
    .
### Answer 1
[answer to question 1]

### Question 2
Which features were involved in this phase? What changes did they undergo? If any
    feature types were modified, answer which features are modified and how they are
    modified. If any features were deleted or created, answer which features are
    deleted or created and provide detailed explanations. (This is a FIXED question
    for each phase.)
### Answer 2
[answer to question 2]

### Question 3
[repeat question 3]
### Answer 3
[answer to question 3]

### Question 4
[repeat question 4]
### Answer 4
[answer to question 4]

### Question 5
[repeat question 5]
### Answer 5
[answer to question 5]

### Question 6
[repeat question 6]
### Answer 6
[answer to question 6]
```

#############
# START REORGANIZE QUESTIONS #
```

## D.2 Unit Tests

In data science competitions, code generated by agents may be executable in the Python interpreter, but this execution does not guarantee correctness. To ensure that data dependencies are properly handled, a Unit Test Tool is necessary. In our research, where the framework operates iteratively, we aim to separate tasks corresponding to different states in data science competitions. Each phase builds upon the results of the previous one, making it crucial to confirm that logic remains sound, data processing is accurate, and information transfers seamlessly from one state to the next. Our Unit Test Tool plays a key role in supporting the self-refine phase of LLM agents.

We developed unit tests (in the accompanying Table 6) based on issues identified during the execution of weak baseline, strong baseline and our AutoKaggle. If the code fails to run in the Python interpreter, an error message is relayed to the agent `Reviewer`. If the code passes this initial stage, it progresses to the Unit Test Tool, where all required tests are executed in a loop. If a test fails, the reason is logged as short-term memory and passed to the next review state. The review and planning stages work in an adversarial interaction: the review phase compiles the reasons for failed unit tests, while the planner addresses these failures in subsequent iterations.

Table 6: Overview of unit tests for state DC, FE, and MBVP. These unit tests handle to detect missing values, outliers, duplicates, and other data consistency issues.

| State | Unit test name | Unit test description |
|---|---|---|
| **State DC** | test_document_exist | Test if cleaned_train.csv and cleaned_test.csv data exist. |
| | test_no_duplicate_cleaned_train | Test if there are any duplicate rows in the cleaned_train.csv. |
| | test_no_duplicate_cleaned_test | Test if there are any duplicate rows in the cleaned_test.csv. |
| | test_readable_cleaned_train | Test if the cleaned_train.csv is readable. |
| | test_readable_cleaned_test | Test if the cleaned_test.csv is readable. |
| | test_cleaned_train_no_missing_values | Test if the cleaned_train.csv contains missing value. |
| | test_cleaned_test_no_missing_values | Test if the cleaned_test.csv contains missing value. |
| | test_cleaned_train_no_duplicated_features | Test if the cleaned_train.csv contains duplicate features. |
| | test_cleaned_test_no_duplicated_features | Test if the cleaned_test.csv contains duplicate features. |
| | test_cleaned_difference_train_test_columns | Test if the cleaned_train.csv and cleaned_test.csv have the same features except for target variable. |
| | test_cleaned_train_no_missing_target | Test if the target variable is in cleaned_train.csv. |
| **State FE** | test_document_exist | Test if processed_train.csv and processed_test.csv data exist. |
| | test_processed_train_feature_number | Test if the feature engineering phase is performed well in processed_train.csv. |
| | test_processed_test_feature_number | Test if the feature engineering phase is performed well in processed_test.csv. |
| | test_file_size | Test if processed data is larger than a threshold. |
| | test_processed_train_no_duplicated_features | Test if the processed_train.csv contains duplicate features. |
| | test_processed_test_no_duplicated_features | Test if the processed_test.csv contains duplicate features. |
| | test_processed_difference_train_test_columns | Test if the processed_train.csv and processed_test.csv have the same features except for target varibale. |
| | test_processed_train_no_missing_target | Test if the target variable is in processed_train.csv. |
| **State MBVP** | test_document_exist | Test if a submission file exists. |
| | test_no_duplicate_submission | Test if there are any duplicate rows in the submission file. |
| | test_readable_submission | test if the submission file is readable. |
| | test_file_num_submission | Test if the submission file and sample_submission.csv have the same number of rows. |
| | test_column_names_submission | Test if the submission file and sample_submission.csv have the same column names. |
| | test_submission_validity | 1) Test if the submission file and sample_submission.csv have the same data index. 2) Test if the submission file and sample_submission.csv have the same numerical range. |

## D.3 MACHINE LEARNING TOOLS DETAILS

Table 7: Overview of Tools for state DC, FE, and MBVP. This table presents various tools categorized by their functionality.

| State | Tool name | Tool description |
|---|---|---|
| | FillMissingValues | Fills missing values or removes columns with missing values based on a threshold. |
| | RemoveColumns WithMissingData | Removes columns containing missing values from a DataFrame based on a threshold. |
| | DetectAndHandleOutliersZscore | Detects and handles outliers in specified columns using the Z-score method. |
| **State DC** | DetectAndHandleOutliersIqr | Detects and handles outliers in specified columns using the Interquartile Range (IQR) method. |
| | RemoveDuplicates | Removes duplicate rows from a DataFrame. |
| | ConvertDataTypes | Converts the data type of specified columns in a DataFrame. |
| | FormatDatetime | Formats datetime columns to a specified format. |
| | OneHotEncode | Performs one-hot encoding on specified categorical columns. |
| | LabelEncode | Performs label encoding on specified categorical columns. |
| | FrequencyEncode | Performs frequency encoding on specified categorical columns. |
| | TargetEncode | Performs target encoding on specified categorical columns. |
| | CorrelationFeatureSelection | Performs feature selection based on correlation analysis. |
| **State FE** | VarianceFeatureSelection | Performs feature selection based on variance analysis. |
| | ScaleFeatures | Scales numerical features in the specified columns of a DataFrame. |
| | PerformPca | Performs Principal Component Analysis (PCA) on the specified columns of a DataFrame. |
| | PerformRfe | Performs Recursive Feature Elimination (RFE) on the specified columns of a DataFrame. |
| | CreatePolynomialFeatures | Creates polynomial features from specified columns of a DataFrame. |
| | CreateFeatureCombinations | Creates feature combinations from specified columns of a DataFrame. |
| **State MBVP** | TrainAndValidation AndSelectTheBestModel | Trains, evaluates, and selects the best machine learning model based on the training data and labels, returning the best performing model along with the performance scores of each model and their best hyperparameters. |

**Examples of Tool Schema.** In this paper, we provide two schema formats for each machine learning tool: JSON and Markdown. Here, we take the FillMissingValues tool as an example and provide schemas in both formats.

---

**Markdown-formatted tool schema for FillMissingValues**

**Description:** Fill missing values in specified columns of a DataFrame. This tool can handle both numerical and categorical features by using different filling methods.
**Applicable Situations:** Handle missing values in various types of features.
**Parameters:**

- **data:**
    - **Type:** `pd.DataFrame`
    - **Description:** A pandas DataFrame object representing the dataset.
- **columns:**
    - **Type:** `string | array`
    - **Description:** The name(s) of the column(s) where missing values should be filled.
- **method:**
    - **Type:** `string`
    - **Description:** The method to use for filling missing values.
    - **Enum:** `auto | mean | median | mode | constant`
    - **Default:** `auto`
- **fill_value:**
    - **Type:** `number | string | null`
    - **Description:** The value to use when method is `constant`.
    - **Default:** `None`

**Required:** `data, columns`
**Result:** Successfully fill missing values in the specified column(s) of data.
**Notes:**

- The `auto` method uses mean for numeric columns and mode for non-numeric columns.
- Using `mean` or `median` on non-numeric columns will raise an error.
- The `mode` method uses the most frequent value, which may not always be appropriate.
- Filling missing values can introduce bias, especially if the data is not missing completely at random.
- Consider the impact of filling missing values on your analysis and model performance.

---

JSON-formatted tool schema for FillMissingValues

```json
{
    "name": "fill_missing_values",
    "description": "Fill missing values in specified columns
of a DataFrame. This tool can handle both numerical and
categorical features by using different filling methods
.",
    "applicable_situations": "handle missing values in
various types of features",
    "parameters": {
        "data": {
            "type": "pd.DataFrame",
            "description": "A pandas DataFrame object
representing the dataset."
        },
        "columns": {
            "type": ["string", "array"],
            "items": {
                "type": "string"
            },
            "description": "The name(s) of the column(s)
where missing values should be filled."
        },
        "method": {
            "type": "string",
            "description": "The method to use for filling
missing values.",
            "enum": ["auto", "mean", "median", "mode", "
constant"],
            "default": "auto"
        },
        "fill_value": {
            "type": ["number", "string", "null"],
            "description": "The value to use when method is '
constant'.",
            "default": null
        }
    },
    "required": ["data", "columns"],
    "result": "Successfully fill missing values in the
specified column(s) of data",
    "additionalProperties": false,
    "notes": [
        "The 'auto' method uses mean for numeric columns and
mode for non-numeric columns.",
        "Using 'mean' or 'median' on non-numeric columns will
 raise an error.",
        "The 'mode' method uses the most frequent value,
which may not always be appropriate.",
        "Filling missing values can introduce bias,
especially if the data is not missing completely at
random.",
        "Consider the impact of filling missing values on
your analysis and model performance."
    ]
}
```

**Tool use.** During execution, we extract the machine learning tools specified in the plan generated by `Planner` and use them as queries to search the entire documentation of machine learning tools. Since the plan includes multiple tools, we retrieve several tools based on their similarity to the queries. The `Developer` then uses the retrieved tools to carry out the task.

## D.4 TOOL UTILIZATION

In the multi-agent framework designed for autonomous data science tasks, tools serve not only as automation resources but also as integral components of the workflow. The framework enables agents to dynamically access and execute tools as they transition through various problem-solving states, ensuring adaptability and efficiency.

The tool utilization process in this framework is structured around a systematic approach. Tool information is first stored in the system's Memory, which is implemented as a vector database. This Memory holds detailed explanations regarding each tool's functionality, usage, and context. A configuration file is used to map specific tools to the states in which they are required, allowing agents to reference and identify the appropriate tools at each stage of the problem-solving process. To determine which tools are required in each state, the table 7 provides an overview of tools categorized by their functionality. As an agent moves into a particular state, it consults the configuration file to determine the relevant tools. From the figure 1 shown, the agent subsequently queries the Memory to retrieve detailed explanations for the tool's use, and finally, executes the tool with precision based on the retrieved information.

This dynamic interaction between the Memory, configuration file, and agents facilitates seamless tool integration, empowering agents to operate autonomously while maintaining flexibility and ensuring accurate tool application throughout the autonomous process.

## D.5 USER INTERACTION

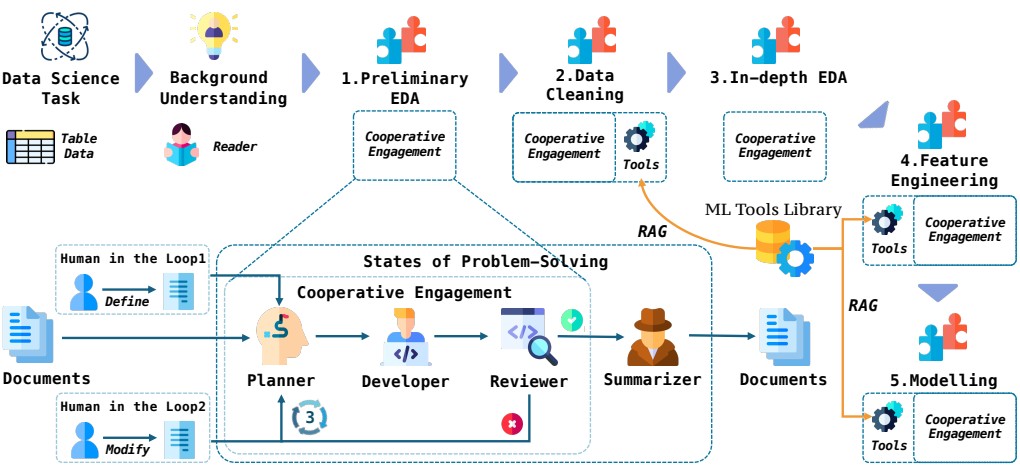

Figure 6: Two human-in-the-loop process.

We have additionally designed two Human-in-the-loop modules for the model. Note that we assessed only the performance of autonomous multi-agents in all of the evaluations we wrote previously, ensuring no human intervention to maintain the fairness and objectivity of our assessment. Figure 6 illustrates these two Human-in-the-Loop methods. Before the `Planner` formulates a plan, human can interact with the command line. The input consists of meticulously manually crafted rules, each one carefully cataloged in a handbook. Memory module subsequently retrieved these predefined rules, integrating this human-driven knowledge in prompt engineering to guide the `Planner`'s next steps. After generating the plan, humans can review and and refine the `Planner`'s output. They inspect areas where the logical flow seems inconsistent, focusing particularly on points where the output diverges from reality to address hallucination issues.

# E  CASE STUDY: TITANIC

## E.1  BACKGROUND UNDERSTANDING

In this step, the system employs a LLM (GPT-4o) to extract and summarize the key information from the Titanic Kaggle competition. Upon completion of this process, a markdown file is automatically generated containing essential competition details, which include the competition overview, dataset information, and evaluation metrics. Below is an excerpt of the generated markdown file:

**Background Understanding**

```
# Competition Information
## 1. Competition Overview
The Titanic competition challenges participants to build a predictive model to
    determine which passengers survived the shipwreck of the RMS Titanic on April 15,
    1912, using passenger data such as name, age, gender, and socio-economic class.

## 2. Files
- **train.csv**: Contains data for 891 passengers, including whether they survived (
    ground truth).
- **test.csv**: Includes data for 418 passengers, but without survival outcomes-these
    need to be predicted.
- **sample_submission.csv**: A template showing the required format for submissions.

## 3. Problem Definition
The objective is to predict the survival of passengers based on various features in
    the dataset. Participants build a model using the training set to learn from the
    data and apply this model to make predictions on the test set.

......

## 6. Evaluation Metrics
The evaluation metric is **accuracy**: the percentage of correct predictions made by
    the model.

## 7. Submission Format
Submissions must be in CSV format with exactly 418 entries plus a header row,
    containing:
- 'PassengerId': Sorted in any order.
- 'Survived': Binary predictions (1 for survived, 0 for deceased).

## 8. Other Key Aspects
- The competition runs indefinitely with a rolling leaderboard.
- Kaggle provides a community forum for discussion and collaboration.
- It's beneficial to explore shared notebooks for additional insights and approaches.
```

## E.2  PRELIMINARY EXPLORATORY DATA ANALYSIS

In this state, an autonomous exploratory analysis is conducted to understand the Titanic dataset. The LLM will plan a process: 1) the dataset is loaded and inspected to reveal its structure, including data types, basic statistics, and missing values. A univariate analysis follows, where the distributions of key numerical features like age and fare are explored, as well as the frequencies of categorical features such as passenger class and gender. 2) A bivariate analysis is performed to examine the relationship between each feature and the target variable, survival. This step involves visualizing the survival rates across different categories and examining how numerical features vary between survivors and non-survivors. 3) A correlation analysis investigates relationships between numerical features, visualized through a correlation matrix and additional pairwise comparisons. This analysis provides a deeper understanding of feature interactions, highlighting those that may have the most significant impact on survival. Below is an excerpt of the generated file:

**Preliminary Exploratory Data Analysis Plan**

```
Great, thank you for the information. Let's move forward with a detailed plan for the
    Preliminary Exploratory Data Analysis phase. The main objectives of this phase are
     to understand the data structure, examine distributions, and identify any initial
     patterns or anomalies. We will break down the task into four key steps:

### 1. Understanding Data Structure
#### Objective:
To get a high-level overview of the data types, missing values, and basic statistics.

#### Actions:
1. **Load the Data**:
   - Use Pandas to read the 'train.csv' and 'test.csv' files.
   - Tools: 'pd.read_csv()'
   - Expected Output: DataFrames for train and test data.
2. **Inspect Data Types and Missing Values**:
   ......
3. **Basic Statistical Summary**:
   ......

### 2. Univariate Analysis of Numerical Features
#### Objective:
To understand the distribution of each numerical feature.

#### Actions:
1. **Histograms for Numerical Features**:
   ......
2. **Box Plots for Numerical Features**:
   ......

### 3. Analysis of Categorical Features
#### Objective:
To examine the distribution and frequency of each categorical feature.

#### Actions:
1. **Bar Plots for Categorical Features**:
   ......
2. **Value Counts**:
   ......

### 4. Initial Insights and Summary
#### Objective:
To summarize the initial findings and identify any immediate data quality issues or
    patterns.

#### Actions:
1. **Summarize Key Findings**:
   ......
2. **Identify Next Steps**:
   ......

### Constraints:
- **Efficiency**: Ensure that visualizations and computations are efficient and
    reproducible.
- **Clarity**: Provide clear and concise visualizations and summaries to aid
    understanding.

### Expected Outputs:
1. DataFrames for train and test sets.
2. Summary of data types and missing values.
3. Basic statistical summaries.
4. Histograms and box plots for numerical features.
5. Bar plots and frequency distributions for categorical features.
6. A detailed summary of initial insights and recommended next steps.

By following this plan, you'll gain a comprehensive preliminary understanding of your
    data, setting a solid foundation for subsequent phases.
```

**Preliminary Exploratory Data Analysis Code**

```python
import os
import pandas as pd
import matplotlib.pyplot as plt
import seaborn as sns

# File paths
train_file_path = '/mnt/d/PythonProjects/AutoKaggleMaster/multi_agents/competition/
    titanic/train.csv'
test_file_path = '/mnt/d/PythonProjects/AutoKaggleMaster/multi_agents/competition/
    titanic/test.csv'

# Loading datasets
train_df = pd.read_csv(train_file_path)
test_df = pd.read_csv(test_file_path)

# Output dataset shapes
print(f"Train dataset shape: {train_df.shape}")
print(f"Test dataset shape: {test_df.shape}")

# Inspect data types and missing values
print("Missing values in train dataset:")
print(train_df.isnull().sum())
print("\nMissing values in test dataset:")
print(test_df.isnull().sum())

# Basic statistical summary for numerical features
print("Statistical summary for numerical features in train dataset:")
print(train_df.describe())

# Plot histograms for numerical features
numerical_features = ['Age', 'Fare', 'SibSp', 'Parch']
for feature in numerical_features:
    plt.figure(figsize=(8, 4))
    sns.histplot(train_df[feature], bins=30, kde=True)
    plt.title(f'Histogram of {feature}')
    plt.xlabel(feature)
    plt.ylabel('Frequency')
    plt.savefig(f'/mnt/d/PythonProjects/AutoKaggleMaster/multi_agents/competition/
     titanic/pre_eda/images/histogram_{feature}.png')
    plt.close()

# Plot bar charts for categorical features
categorical_features = ['Pclass', 'Sex', 'Embarked', 'Survived']
for feature in categorical_features:
    plt.figure(figsize=(8, 4))
    sns.countplot(x=train_df[feature])
    plt.title(f'Bar chart of {feature}')
    plt.xlabel(feature)
    plt.ylabel('Count')
    plt.savefig(f'/mnt/d/PythonProjects/AutoKaggleMaster/multi_agents/competition/
     titanic/pre_eda/images/bar_chart_{feature}.png')
    plt.close()

# Summarize key findings
key_findings = """
Key Findings from Preliminary EDA:
1. Missing Values:
   - Age has 177 missing values in the train dataset.
   - Cabin has significant missing values.
   - Embarked has 2 missing values in the train dataset.

2. Numerical Features:
   - Age distribution shows a right skew.
   - Fare distribution is highly skewed.

3. Categorical Features:
   - More males than females in the train dataset.
   - Most passengers embarked from port 'S' (Southampton).
   - Survival rate is around 38% in the train dataset.
"""
print(key_findings)
```

**Preliminary EDA**

```
Train Dataset Head:
    PassengerId  Survived  Pclass  ...     Fare Cabin  Embarked
0             1         0       3  ...   7.2500   NaN         S
1             2         1       1  ...  71.2833   C85         C
2             3         1       3  ...   7.9250   NaN         S
3             4         1       1  ...  53.1000  C123         S
4             5         0       3  ...   8.0500   NaN         S

[5 rows x 12 columns]

Test Dataset Head:
    PassengerId  Pclass  ... Cabin Embarked
0           892       3  ...   NaN        Q
1           893       3  ...   NaN        S
2           894       2  ...   NaN        Q
3           895       3  ...   NaN        S
4           896       3  ...   NaN        S

[5 rows x 11 columns]

......

Missing Values in Test Dataset:
 PassengerId     0
Pclass          0
Name            0
Sex             0
Age            86
SibSp           0
Parch           0
Ticket          0
Fare            1
Cabin         327
Embarked        0
dtype: int64

Frequency of Pclass:
 Pclass
3    491
1    216
2    184
Name: count, dtype: int64

Frequency of Sex:
 Sex
male      577
female    314
Name: count, dtype: int64

Frequency of Embarked:
 Embarked
S    644
C    168
Q     77
Name: count, dtype: int64
```

## E.3 DATA CLEANING

We demonstrate the data analysis capabilities of our framework using the age column from the Titanic competition's training set as an example. In the pre-EDA phase, the distribution of the age histogram is as shown in Figure 7. During the data cleaning phase, we filter out missing values using unit tests. You can see a comparison of the age box plots before and after the outliers have been processed in Figure 8. In the deep-EDA phase, the distribution of the age histogram is as shown in Figure 9.

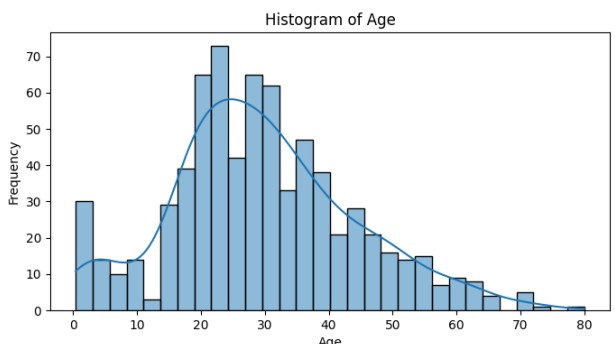

Figure 7: The histogram of age before outliers are processed

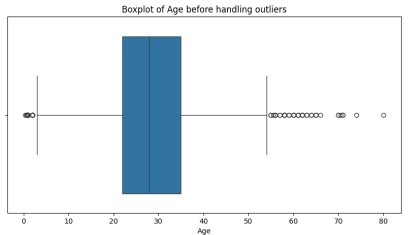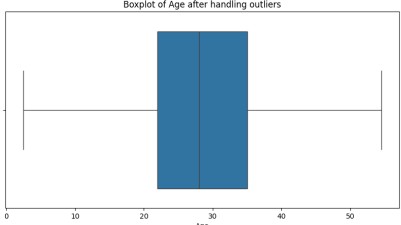

Figure 8: Comparison of age box plots before and after treatment of outliers.The image on the left is before the outliers are processed and the one on the right is after the process is done

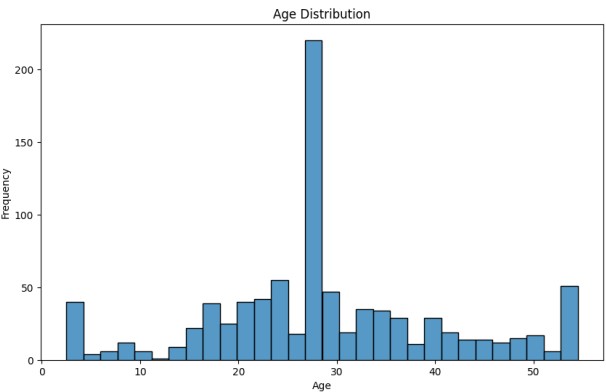

Figure 9: The histogram of age after outliers are processed

**Data Cleaning Code**

```python
import sys
import os
import numpy as np
import pandas as pd
import matplotlib.pyplot as plt
import seaborn as sns

sys.path.extend(['.', '..', '../..', '../../..', '../../../..', 'multi_agents', '
    multi_agents/tools', 'multi_agents/prompts'])
sys.path.append(os.path.dirname(os.path.abspath(__file__)))
from tools.ml_tools import *

def generated_code_function():
    # Load datasets
    train_path = '/mnt/d/PythonProjects/AutoKaggleMaster/multi_agents/competition/
     titanic/train.csv'
    test_path = '/mnt/d/PythonProjects/AutoKaggleMaster/multi_agents/competition/
     titanic/test.csv'
    train_df = pd.read_csv(train_path)
    test_df = pd.read_csv(test_path)

    # Handle missing values
    for df in [train_df, test_df]:
        df = fill_missing_values(df, columns=['Age', 'Fare'], method='median')
        df = fill_missing_values(df, columns=['Embarked'], method='mode')
        df = remove_columns_with_missing_data(df, columns=['Cabin'])

    # Convert data types
    for df in [train_df, test_df]:
        df = convert_data_types(df, columns=['PassengerId', 'Pclass'], target_type='
     str')
    train_df = convert_data_types(train_df, columns=['Survived'], target_type='str')

    # Plot outliers and handle using IQR method
    def plot_outliers(data, columns, suffix):
        output_dir = '/mnt/d/PythonProjects/AutoKaggleMaster/multi_agents/competition/
     titanic/data_cleaning/images/'
        os.makedirs(output_dir, exist_ok=True)
        for column in columns:
            plt.figure(figsize=(10, 5))
            sns.boxplot(x=data[column])
            plt.title(f'Boxplot of {column} {suffix}')
            plt.savefig(f'{output_dir}{column}_{suffix}.png')
            plt.close()

    columns_with_outliers = ['Age', 'Fare']
    plot_outliers(train_df, columns_with_outliers, 'before_outliers')

    for df in [train_df, test_df]:
        df = detect_and_handle_outliers_iqr(df, columns=columns_with_outliers, factor
     =1.5, method='clip')

    plot_outliers(train_df, columns_with_outliers, 'after_outliers')

    # Save cleaned datasets
    train_df.to_csv('/mnt/d/PythonProjects/AutoKaggleMaster/multi_agents/competition/
     titanic/cleaned_train.csv', index=False)
    test_df.to_csv('/mnt/d/PythonProjects/AutoKaggleMaster/multi_agents/competition/
     titanic/cleaned_test.csv', index=False)

if __name__ == "__main__":
    generated_code_function()
```

**Data Cleaning Result**

```
Missing values in train dataset before handling:
PassengerId      0
...
Sex              0
Age            177
...
Cabin          687
Embarked         2
dtype: int64

Missing values in test dataset before handling:
PassengerId      0
...
Age             86
...
Fare             1
Cabin          327
Embarked         0
dtype: int64

Missing values in train dataset after handling:
Age            0
Embarked       0
...
...
SibSp          0
Ticket         0
dtype: int64

Missing values in test dataset after handling:
Age            0
Embarked       0
...
...
SibSp          0
Ticket         0
dtype: int64

Data types in train dataset after conversion:
Age           float64
Embarked       object
Fare          float64
Name           object
Parch           int64
PassengerId    object
Pclass         object
Sex            object
SibSp           int64
Survived       object
Ticket         object
dtype: object

Data types in test dataset after conversion:
Age           float64
Embarked       object
Fare          float64
Name           object
Parch           int64
PassengerId    object
Pclass         object
Sex            object
SibSp           int64
Ticket         object
dtype: object
Cleaned training data saved to /mnt/d/PythonProjects/AutoKaggleMaster/multi_agents/
    competition/titanic/cleaned_train.csv
Cleaned test data saved to /mnt/d/PythonProjects/AutoKaggleMaster/multi_agents/
    competition/titanic/cleaned_test.csv
```

## E.4  IN-DEPTH EXPLORATORY DATA ANALYSIS

In this state, the AutoKaggle delves further into the Titanic dataset. 1) The process begins with an extended univariate analysis to explore the distribution of both numerical and categorical features.

Key statistical summaries are generated for numerical features such as age and fare, while bar charts and frequency tables are created for categorical features like passenger class and gender. 2) A bivariate analysis investigates the relationship between individual features and the survival outcome. Box plots and violin plots are used to analyze how numerical features vary between survivors and non-survivors, while count plots are generated for categorical features to visualize survival rates across different groups. 3) A correlation analysis is conducted to explore the relationships between numerical features, visualized through a correlation matrix and heatmap. This helps to identify any strong correlations between features and the target variable, survival. 4) A multivariate analysis is performed to explore interactions between key features such as passenger class, gender, and age, in relation to survival. Visualizations, such as stacked bar charts and facet grids, are used to highlight these complex interactions, providing deeper insights into the data.Below is an excerpt of the generated file:

1944
1945
1946
1947
1948
1949
1950
1951
1952
1953
1954
1955
1956
1957
1958
1959
1960
1961
1962
1963
1964
1965
1966
1967
1968
1969
1970
1971
1972
1973
1974
1975
1976
1977
1978
1979
1980
1981
1982
1983
1984
1985
1986
1987
1988
1989
1990
1991
1992
1993
1994
1995
1996
1997

**In-depth Exploratory Data Analysis Plan**

```
Great! Based on the information provided, we will proceed with a detailed plan for the
    **In-depth Exploratory Data Analysis (EDA)** phase. The aim is to perform a
    thorough analysis of each feature to uncover deeper insights that can inform
    feature engineering and model building.

### Plan for In-Depth Exploratory Data Analysis

#### Task 1: Detailed Univariate Analysis
**Objective:** Understand the distribution and characteristics of each individual
    feature.

1. **Age:**
   - **Actions:**
     - Plot a histogram to observe the distribution.
     - Use a boxplot to identify any remaining outliers.
   - **Tools & Parameters:**
     - 'seaborn.histplot(data=cleaned_train, x='Age', bins=30)'
     - 'seaborn.boxplot(data=cleaned_train, x='Age')'
   - **Expected Output:**
     - Histogram showing the age distribution.
     - Boxplot indicating the presence of outliers.
   - **Constraints:**
     - Ensure plots are clear and readable.

2. **Fare:**
   ......

3. **Categorical Features (Pclass, Sex, Embarked):**
   ......

#### Task 2: Bivariate Analysis
**Objective:** Explore relationships between pairs of features, especially with the
    target variable 'Survived'.

1. **Survived vs. Age:**
   ......

2. **Survived vs. Fare:**
   ......

3. **Survived vs. Categorical Features (Pclass, Sex, Embarked):**
   ......

#### Task 3: Multivariate Analysis
**Objective:** Understand interactions between multiple features and their combined
    effect on the target variable.

1. **Pairplot:**
   ......

2. **Correlation Heatmap:**
   ......

#### Task 4: Insights Summary
**Objective:** Summarize key findings from the in-depth EDA to inform the next phases.

1. **Actions:**
   ......
2. **Tools & Parameters:**
   ......
3. **Expected Output:**
   ......
4. **Constraints:**
   ......

By following this detailed plan, you will be able to gain a thorough understanding of
    the dataset, which will be invaluable for the subsequent phases of feature
    engineering and model building.
```

**In-depth Exploratory Data Analysis Code**

```python
import pandas as pd
import matplotlib.pyplot as plt
import numpy as np

# Load the cleaned training data
cleaned_train = pd.read_csv('cleaned_train.csv')
# Task 1: Univariate analysis on the Age and Fare features

# Histogram and Boxplot for Age and Fare
for feature in ['Age', 'Fare']:
    plt.figure(figsize=(10, 6))
    cleaned_train[feature].hist(bins=30)
    plt.title(f'{feature} Distribution')
    plt.xlabel(feature)
    plt.ylabel('Frequency')
    plt.savefig(f'deep_eda/images/{feature.lower()}_histogram.png')
    plt.close()
    plt.figure(figsize=(10, 6))
    cleaned_train[feature].plot(kind='box')
    plt.title(f'Boxplot of {feature}')
    plt.xlabel(feature)
    plt.savefig(f'deep_eda/images/{feature.lower()}_boxplot.png')
    plt.close()

print("Univariate analysis on Age and Fare features completed.")

# Task 2: Univariate analysis on categorical features
# Countplot for Pclass, Sex, Embarked
for feature in ['Pclass', 'Sex', 'Embarked']:
    cleaned_train[feature].value_counts().plot(kind='bar', figsize=(10, 6), title=f'{
     feature} Distribution')
    plt.xlabel(feature)
    plt.ylabel('Frequency')
    plt.savefig(f'deep_eda/images/{feature.lower()}_countplot.png')
    plt.close()

# Task 3: Bivariate analysis of Survived vs. Age, Fare, categorical features
# Violin plot for Age and Fare by Survived
for feature in ['Age', 'Fare']:
    plt.figure(figsize=(10, 6))
    cleaned_train.boxplot(column=feature, by='Survived')
    plt.title(f'Survival by {feature}')
    plt.xlabel('Survived')
    plt.ylabel(feature)
    plt.savefig(f'deep_eda/images/survived_vs_{feature.lower()}.png')
    plt.close()

# Countplot for categorical features by Survived
for feature in ['Pclass', 'Sex', 'Embarked']:
    pd.crosstab(cleaned_train[feature], cleaned_train['Survived']).plot(kind='bar',
     stacked=True, figsize=(10, 6))
    plt.title(f'Survival by {feature}')
    plt.xlabel(feature)
    plt.ylabel('Count')
    plt.savefig(f'deep_eda/images/survived_vs_{feature.lower()}.png')
    plt.close()

# Task 4: Multivariate analysis using a correlation heatmap
# Correlation heatmap
numeric_df = cleaned_train.select_dtypes(include=[np.number])
plt.figure(figsize=(10, 8))
plt.matshow(numeric_df.corr(), cmap='coolwarm', fignum=1)
plt.title('Correlation Heatmap')
plt.savefig('deep_eda/images/correlation_heatmap.png')
plt.close()

# Task 5: Summarize key insights from the EDA
summary = """
......
"""
# Save the summary to a text file
with open('deep_eda/eda_summary.txt', 'w') as file:
    file.write(summary)
```

**In-deep EDA**

```
Summary statistics for Age:
 count    891.000000
mean      29.039282
std       12.072074
min        2.500000
25%       22.000000
50%       28.000000
75%       35.000000
max       54.500000
Name: Age, dtype: float64

......

Survival rate by Pclass:
 Pclass
1    0.629630
2    0.472826
3    0.242363
Name: Survived, dtype: float64
Survival rate by Sex:
 Sex
female    0.742038
male      0.188908
Name: Survived, dtype: float64
Survival rate by Embarked:
 Embarked
C    0.553571
Q    0.389610
S    0.339009
Name: Survived, dtype: float64
Correlation matrix:
                Age      SibSp     Parch      Fare  Survived
Age        1.000000 -0.239601 -0.178959  0.144544 -0.060622
SibSp     -0.239601  1.000000  0.414838  0.332021 -0.035322
Parch     -0.178959  0.414838  1.000000  0.292616  0.081629
Fare       0.144544  0.332021  0.292616  1.000000  0.317430
Survived  -0.060622 -0.035322  0.081629  0.317430  1.000000
```

### E.5 FEATURE ENGINEERING

In this phase, the AutoKaggle add several new features to enhance the predictive power of the dataset. 1) A FamilySize feature is created by combining the SibSp and Parch columns, representing the total number of family members aboard, including the passenger. This feature captures the familial context, which could influence survival likelihood. 2) An AgeGroup feature is derived by categorizing passengers into age groups, simplifying the continuous age variable into meaningful categories such as "Child" and "Senior." This transformation helps identify potential age-related survival patterns. 3) Categorical features like Sex, Embarked, and Pclass are then encoded into numerical form to ensure they can be used in the model. One-hot encoding is applied to Sex and Embarked, while label encoding is used for Pclass, respecting its ordinal nature. 4) The cabin data is processed by extracting the first letter of the Cabin feature to create a new Deck variable. This feature provides information about the passenger's location on the ship, which may correlate with survival outcomes. Missing cabin data is handled by assigning an 'Unknown' category, ensuring completeness of the feature.

**Feature Engineering Plan**

```
Great! Let's design a detailed plan for the Feature Engineering phase, focusing on the
    current features and the available tools. We'll ensure that each task is clear,
    specific, and includes the necessary steps to achieve the desired outcome.

### Feature Engineering Plan

#### Task 1: Create New Features
**Objective:** Enhance the dataset by creating new features that could capture
    additional information relevant to predicting survival.

1. **Family Size:**
   - **Action:** Create a new feature 'FamilySize' by combining 'SibSp' and 'Parch'.
   - **Method:** 'FamilySize = SibSp + Parch + 1' (including the passenger themselves)
     .
   - **Impact:** Captures the total number of family members traveling together, which
      can influence survival chances.

2. **IsAlone:**
   ......

3. **Age Bins:**
   ......

4. **Fare per Person:**
   ......

#### Task 2: Encode Categorical Features
**Objective:** Convert categorical features into numerical format suitable for machine
    learning models.

1. **Sex:**
   ......

2. **Embarked:**
   ......

#### Task 3: Handle Ticket and Cabin Features
**Objective:** Extract useful information from 'Ticket' and 'Cabin' features, which
    are currently in text format.

1. **Ticket:**
   ......

2. **Cabin:**
   ......

#### Task 4: Scale Numerical Features
**Objective:** Standardize numerical features to ensure they are on a comparable scale
    , improving model performance.

1. **Numerical Features:**
   ......

### Summary of Expected Output
- **New Features Added:** 'FamilySize', 'IsAlone', 'AgeBins', 'FarePerPerson'
- **Encoded Features:** 'Sex' (label encoded), 'Embarked' (one-hot encoded)
- **Processed Features:** 'TicketPrefix', 'CabinDeck'
- **Scaled Features:** 'Age', 'Fare', 'FarePerPerson'

### Constraints and Considerations
- **Runtime Efficiency:** Ensure feature creation and encoding steps are optimized for
     performance.
- **Handling Missing Values:** Address any missing values in the newly created
    features appropriately.
- **Avoid Data Leakage:** Perform encoding and scaling separately on train and test
    sets to prevent data leakage.

By following these tasks, you will transform the cleaned data into a more informative
    and model-ready format ('processed_train.csv' and 'processed_test.csv'). This plan
     ensures that the most critical steps of feature engineering are covered,
    enhancing the predictive power of your model for the Titanic competition.
```

**Feature Engineering Code**

```python
import pandas as pd

# Load the cleaned datasets
train_df = pd.read_csv('cleaned_train.csv')
test_df = pd.read_csv('cleaned_test.csv')

# Create copies of the dataframes
train_df_copy = train_df.copy()
test_df_copy = test_df.copy()

# STEP 1: Create New Features
train_df_copy['FamilySize'] = train_df_copy['SibSp'] + train_df_copy['Parch'] + 1
test_df_copy['FamilySize'] = test_df_copy['SibSp'] + test_df_copy['Parch'] + 1

train_df_copy['IsAlone'] = (train_df_copy['FamilySize'] == 1).astype(int)
test_df_copy['IsAlone'] = (test_df_copy['FamilySize'] == 1).astype(int)

age_bins = [0, 12, 18, 35, 60, 120]
age_labels = ['Child', 'Teen', 'Adult', 'Senior', 'Elder']
train_df_copy['AgeBins'] = pd.cut(train_df_copy['Age'], bins=age_bins, labels=
    age_labels, right=False)
test_df_copy['AgeBins'] = pd.cut(test_df_copy['Age'], bins=age_bins, labels=age_labels
    , right=False)

train_df_copy['FarePerPerson'] = train_df_copy['Fare'] / train_df_copy['FamilySize']
test_df_copy['FarePerPerson'] = test_df_copy['Fare'] / test_df_copy['FamilySize']

# Save the datasets with new features
train_df_copy.to_csv('processed_train.csv', index=False)
test_df_copy.to_csv('processed_test.csv', index=False)

# Load the processed datasets
train_df = pd.read_csv('processed_train.csv')
test_df = pd.read_csv('processed_test.csv')

# Define functions to extract ticket prefix and cabin deck
def extract_ticket_prefix(ticket):
    parts = ticket.split()
    return parts[0] if not parts[0].isdigit() else 'None'

def extract_cabin_deck(cabin):
    return cabin[0] if pd.notna(cabin) else 'Unknown'

# Extract TicketPrefix and CabinDeck
train_df['TicketPrefix'] = train_df['Ticket'].apply(extract_ticket_prefix)
test_df['TicketPrefix'] = test_df['Ticket'].apply(extract_ticket_prefix)

train_df['CabinDeck'] = train_df['Cabin'].apply(extract_cabin_deck) if 'Cabin' in
    train_df.columns else 'Unknown'
test_df['CabinDeck'] = test_df['Cabin'].apply(extract_cabin_deck) if 'Cabin' in
    test_df.columns else 'Unknown'

# Save the datasets with extracted features
train_df.to_csv('processed_train.csv', index=False)
test_df.to_csv('processed_test.csv', index=False)
```

### E.6   MODEL BUILDING, VALIDATION, AND PREDICTION

In this phase, we conduct a comprehensive analysis of the Titanic passenger dataset with the aim of predicting passengers' survival probabilities. Initially, the data undergo preprocessing that included filling missing values, deleting columns with high missingness, and handling outliers. Subsequent feature engineering efforts introduce new attributes such as family size, solitary travel, age groupings, and fare per person, and involved encoding for gender and embarkation points. Furthermore, a random forest model is employed, optimized via grid search, and evaluated using cross-validation. Predictions are then made on the test set, and a submission file is prepared.

**Model Building, Validation, and Prediction Plan**

```
### Detailed Plan

#### Task 1: Prepare Training Data
**Objective:** Separate the target variable and remove non-numeric columns.

1. **Separate Target Column**
   - **Action:** Extract the `Survived` column from `processed_train.csv` as `y`.
   - **Tool:** pandas
   - **Expected Output:** `y` as a separate pandas Series containing the target
     variable.
   - **Code Example:**
     ```python
     import pandas as pd
     train_data = pd.read_csv('/mnt/d/PythonProjects/AutoKaggleMaster/multi_agents/
     competition/titanic/processed_train.csv')
     y = train_data['Survived']
     ```

2. **Remove Non-Numeric Columns**
   - **Action:** Identify and drop non-numeric columns from the training set.
   - **Features Involved:** `Name`, `Ticket`, `TicketPrefix`, `CabinDeck`
   - **Tool:** pandas
   - **Expected Output:** `X_train` as a DataFrame containing only numeric columns.
   - **Code Example:**
     ```python
     X_train = train_data.drop(columns=['Survived', 'Name', 'Ticket', 'TicketPrefix',
     'CabinDeck'])
     ```

#### Task 2: Prepare Test Data
**Objective:** Ensure the test data has the same structure as the training data.

1. **Remove Non-Numeric Columns**
   ......

#### Task 3: Train and Validate the Model
**Objective:** Train and validate a machine learning model using the prepared datasets
     .

1. **Model Selection and Training**
   ......

2. **Hyperparameter Tuning**
   ......

#### Task 4: Make Predictions
**Objective:** Use the trained model to make predictions on the test dataset.

1. **Prediction**
   ......

#### Task 5: Prepare Submission File
**Objective:** Prepare the submission file in the required format.

1. **Submission File Preparation**
   ......

### Final Notes
- Ensure that all steps are executed in sequence to maintain data integrity.
- Validate each step with print statements or logs to confirm successful execution.
- Use cross-validation to ensure model robustness and avoid overfitting.

Feel free to execute the tasks based on this detailed plan. If you have any questions
     or need further adjustments, don't hesitate to ask.
```

**Model Building, Validation, and Prediction Code**

```python
import pandas as pd
# Load the processed training data
processed_train_path = '/mnt/d/PythonProjects/AutoKaggleMaster/multi_agents/
    competition/titanic/processed_train.csv'
train_df = pd.read_csv(processed_train_path)

# Separate the target variable
y = train_df['Survived']

# Identify non-numeric columns to be dropped
columns_to_drop = ['Survived', 'Name', 'Ticket', 'TicketPrefix', 'CabinDeck', 'AgeBins
    ']

# Drop the target variable and non-numeric columns
X_train = train_df.drop(columns=columns_to_drop)

# Load the processed test data
processed_test_path = '/mnt/d/PythonProjects/AutoKaggleMaster/multi_agents/competition
    /titanic/processed_test.csv'
test_df = pd.read_csv(processed_test_path)

# Drop non-numeric columns to match X_train
X_test = test_df.drop(columns=['Name', 'Ticket', 'TicketPrefix', 'CabinDeck', 'AgeBins
    '])

# Ensure the column order matches X_train
X_test = X_test[X_train.columns]

print(f"Test data processed. X_test shape: {X_test.shape}")

from sklearn.ensemble import RandomForestClassifier
from sklearn.model_selection import GridSearchCV, cross_val_score

# Define the model and parameter grid
model = RandomForestClassifier(random_state=42)
param_grid = {
    'n_estimators': [100, 200, 300],
    'max_depth': [None, 10, 20, 30]
}

# Use GridSearchCV for hyperparameter tuning
grid_search = GridSearchCV(model, param_grid, cv=5, scoring='accuracy')
grid_search.fit(X_train, y)

# Get the best model
best_model = grid_search.best_estimator_

# Validate the model using cross-validation
cv_scores = cross_val_score(best_model, X_train, y, cv=5, scoring='accuracy')

# Make predictions on the test data
predictions = best_model.predict(X_test)

# Convert predictions to a DataFrame
predictions_df = pd.DataFrame({
    'PassengerId': test_df['PassengerId'],
    'Survived': predictions
})

print(f"Predictions made. Sample predictions:\n{predictions_df.head()}")

# Prepare the submission file
submission_path = '/mnt/d/PythonProjects/AutoKaggleMaster/multi_agents/competition/
    titanic/submission.csv'
predictions_df.to_csv(submission_path, index=False)
```

### E.7 RESEARCH REPORT

After completing the entire data science pipeline, AutoKaggle reviews the interim reports from each phase and finally compiles a Research Report to document its key findings, major decisions, and their rationale at every phase. This helps users better understand the solutions provided by AutoKaggle.

---

**Research Report**

```
# COMPETITION RESEARCH REPORT

## 1. PRELIMINARY EDA

The preliminary exploratory data analysis (EDA) involved examining the dataset's
    structure and identifying key features. The analysis revealed critical insights
    such as:
- **Missing Values**: Notably, the 'Age' and 'Cabin' features had significant missing
    values.
- **Outliers**: Outliers were identified in the 'Fare' and 'Age' features, with fares
    exceeding 200 and ages above 60.
- **Statistical Summary**: The median age was around 30, while the median fare was
    approximately 14.45.
...

## 2. DATA CLEANING

The data cleaning phase focused on addressing missing values and outliers. Key actions
    included:
- **Imputation**:
  - 'Age': Missing values were filled with the mean.
  - 'Cabin': Filled with 'Missing' to indicate absence.
  - 'Embarked': Missing values filled with the mode.
- **Outlier Treatment**:
  - 'Fare' and 'Age' were clipped using the IQR method to reduce the influence of
    outliers.
...

## 3. DEEP EDA

In-depth EDA revealed valuable insights about the relationships between features and
    survival rates:
- **Pclass**: Higher classes showed higher survival rates; average Pclass was 2.31.
- **Sex**: Females had a significantly higher survival rate compared to males.
- **Fare**: Higher fares correlated positively with survival, indicating a potential
    link to better accommodation and safety.
...

## 4. FEATURE ENGINEERING

Feature engineering involved creating and transforming features to enhance model
    performance:
- **New Features Created**:
  - **Title**: Extracted from 'Name'.
  - **FamilySize**: Summation of 'SibSp' and 'Parch'.
  - **IsAlone**: Binary feature indicating if a passenger traveled alone.
  - **FarePerPerson**: Calculated fare per individual in a family.
...

## 5. MODEL BUILDING, VALIDATION, AND PREDICTION

Multiple models were trained during this phase, including:
- **Models**: XGBoost, SVM, Random Forest, Decision Tree, and Logistic Regression.
- **Best Model**: Random Forest achieved the highest validation score of 0.8379.
...

## 6. CONCLUSION

The competition's approach involved a structured process of EDA, data cleaning,
    feature engineering, and model evaluation. Key insights included the strong
    influence of 'Sex', 'Pclass', and 'Fare' on survival rates. The most impactful
    decisions involved addressing missing values and outliers, which collectively
    improved data quality and model accuracy. Future recommendations include further
    feature engineering, hyperparameter tuning, and validation of feature importance
    to enhance model performance.
```

