# OpenReview forum: "AutoKaggle: A Multi-Agent Framework for Autonomous Data Science Competitions"
_ICLR.cc/2025/Conference — Submitted to ICLR 2025_

### Official Review · Reviewer_u3cH · 2024-11-02

**Soundness:** 2
**Presentation:** 3
**Contribution:** 3
**Rating:** 5
**Confidence:** 4

**Summary:**

This paper presents a scaffolding framework which uses LLMs to create a multi-agent system, used to attempt Kaggle problems. They use a "phase-based" multi-agent approach, together with a library of hand-crafted ML tools, and extensive hand-crafted unit-tests tailored to the Kaggle problems.

Applying this framework to 8 Kaggle problems (4 pre-GPT-4 training cut-off, 4 afterwards), they achieve a significant solve rate, and an average of 42% on the Kaggle leaderboard.

The paper also explores ablation of various modules (various tools, and the unit-testing module).

**Strengths:**

A system which can score 43% on Kaggle leaderboards is a significant milestone on the path to automated coding and datascience. Additionally, since many challenges which such a system would face would also arise in more general task-completion (e.g. long-term planning, establishing coherency, managing context, preventing execution from looping) and so would transfer to improve AI agents in general.

Great collection of Classic and Recent challenges, and baselines seem reasonable (though see my Q about the Strong Baseline).

It's helpful to have this variety of scores (though see my Q about CS).

Architecture is clearly laid out, and the paper is overall very easy to read.

Clear exploration and explanation of the underlyring readon why the feature-engineering tools reduce the framework's score (many features, leading to more complexity than the agents can handle).

**Weaknesses:**

Whenever CoT is used as an interpretability tool, I think it's always wise to mention unfaithfulness e.g. https://arxiv.org/abs/2305.04388

There are two places where a long list is hard to read:
#1 ~L78: AutoKaggle integrates a comprehensive machine learning tools library, covering three core toolsets: data cleaning, feature engineering, and model building, validation, and prediction.

#2 ~L186: The data science process is divided into six key stages: understanding the
background, preliminary exploratory data analysis, data cleaning, in-depth exploratory data anal-
ysis, feature engineering, and model building, validation, and prediction

Perhaps "model-building, -validation, and -prediction" would be easier to read.

~L146: I'm surprised not to see mentioned what seems to me to be the main thing underlying the motivation of multi-agent systems: finite context length, requiring summarisation and specialisation.

It's not clear how much of the headline 43% on the leaderboard is down to the skill of the human-in-the-loop, which severely undermines the claim. Without a comparison to how well the human takes unassisted (in terms of success rate or time taken), or to how well AutoKaggle performs without HITL, it's impossible to reliably state how effective the framework is.

Unspecified HITL also undermines the various claims of a "fully automated framework" (e.g. L175)

Not much detail on these unit tests. Who writes them? What's the coverage like? Are there any guarantees? If (as I suspect) the "meticulously designed" unit tests are written by humans, then we have a similar situation as with the unspecified human-in-the-loop: the framework is not "fully automated", and it's impossible to rigorously determine how much effect the human hand-holding has on the framework's suggess. This should, at minimum, be clearly, explicitly and boldly acknowledged.

Additionally, it is unclear to me how much of the ML-tools library was developed alongside particular Kaggle Competition attempts. If the tools were developed on a case-by-case basis, to address hurdles found in the challenge, then there is significant data leakage from the evaluation dataset to the framework, leading to overfitting to the competitions chosen during development, and much of the headline 43% comes from tools handcrafted by human developers on a case-by-case basis. For a fair validation of how well this framework performs in "fully automated" mode, the library would need to be "frozen" while the framework was tested on a held-out set of Kaggle Competitions.

Very minor point: ~L350, I agree that there is a risk of data leakage for competitions from before Oct '23, however to say that GPT-4o's training data includes Classic Kaggle is an assumption: better to say simply that there is a risk of data leakage.

If you're considering data leakage, it would be worth flagging that the 42% includes Classic problems: using only the newer problems, performance is slightly below human average.

**Questions:**

~L140, you say that CoT improves reasoning at the expense of introducing hallucinations. Is there any evidence that CoT makes models any more or less likely to hallucinate?

~L141, you say that the ReAct paradigm addresses hallucinations - that's not my understanding of what ReAct does or how it works, my understanding is that it combines thoughts and actions, yes, but that this has nothing to do with hallucinations or refining outputs.

~L360: What is the difference between "Success - Non-compliant" and "Success - Compliant"?

~L403: What's the justification / motivation for the complex / compound "Comprehensive Score"? How does it compare to other measures, what specifically does it achieve or avoid?

~L431: could you say more about this "strong baseline" - I don't understand its construction.

If adding FE tools drops performance because FE adds too much complexity, then why does "All tools" (which presumably includes FE tools) recover this performance?

---

> ### Author Response · Authors · 2024-11-24
> **Thanks for your valuable comments! Authors' feedback [1/4]**
>
> Thanks for your valuable feedback and constructive comments. Below, we present our point-by-point response to the weaknesses and comments identified in our submission:
>
> > **W1:** Whenever CoT is used as an interpretability tool, I think it's always wise to mention unfaithfulness e.g. https://arxiv.org/abs/2305.04388
>
> Thanks for your suggestion, we have already cited it in our paper.
>
> ---
>
> > **W2:** There are two places where a long list is hard to read: #1 ~L78: AutoKaggle integrates a comprehensive machine learning tools library, covering three core toolsets: data cleaning, feature engineering, and model building, validation, and prediction.
> >
> > \#2 ~L186: The data science process is divided into six key stages: understanding the background, preliminary exploratory data analysis, data cleaning, in-depth exploratory data anal- ysis, feature engineering, and model building, validation, and prediction
> >
> > Perhaps "model-building, -validation, and -prediction" would be easier to read.
>
> Thanks for pointing out the issues. We have optimized these expressions in the new version of the paper.
>
> ---
>
> > **W3:** ~L146: I'm surprised not to see mentioned what seems to me to be the main thing underlying the motivation of multi-agent systems: finite context length, requiring summarisation and specialisation.
>
> Thanks for your valuable feedback, and this is a great question. Regarding the issue of limited context length mentioned, we have implemented the following optimizations in AutoKaggle:
>
> In our framework, the primary agents for each stage include the Planner, Developer, Reviewer, and Summarizer. Taking the Planner as an example, it receives the following inputs:
>
> 1. **Report from the previous stage**: This is a summary generated by the Summarizer from the preceding stage, covering changes in files, variations in data features, and key findings from the previous phase. The average length of this input is approximately 1,872 tokens.
> 2. **Tool documentation**: We employ a retrieval-augmented generation (RAG) approach to extract only the necessary tool information from the overall documentation, limiting the length to within 4,996 tokens.
> 3. **Plan from the previous stage**: This input ensures consistency in data processing logic across stages and has an average length of about 1,453 tokens.
> 4. **The Planner's prompt information**: This is specific to the Planner and has an average length of 1,176 tokens.
>
> (The above averages are based on five iterations of AutoKaggle completing Task 1 of the Titanic competition.)
>
> In summary, the total input length for the Planner is approximately 9,497 tokens, which is well within 1/10 of GPT-4o's context window of 128,000 tokens. Therefore, the unique compression of the input information is unnecessary. Furthermore, our design highlights careful information summarization and thoughtful input allocation to each agent in the framework.
>
> ---
>
> > **W4:** It's not clear how much of the headline 43% on the leaderboard is down to the skill of the human-in-the-loop, which severely undermines the claim. Without a comparison to how well the human takes unassisted (in terms of success rate or time taken), or to how well AutoKaggle performs without HITL, it's impossible to reliably state how effective the framework is.
> >
> > Unspecified HITL also undermines the various claims of a "fully automated framework" (e.g. L175)
>
> Thanks for your valuable feedback. We apologize for the confusion in our writing and figures. It is essential to clarify that the test results presented in the paper were achieved without any human intervention. One key feature of our framework is that it allows for manual adjustments. Users can modify parameters in the config.json file to enable customization at each stage. For example, users can manually revise the plan to better align with specific requirements or goals after the Planner completes its planning phase. This feature provides flexibility for tailoring the framework to diverse scenarios. We plan to conduct a user study where independent users will try AutoKaggle in their daily data science scenarios. They will provide feedback on the ease of setup and interaction.

---

> ### Author Response · Authors · 2024-11-24
> **Thanks for your valuable comments! Authors' feedback [2/4]**
>
> > **W5:** Not much detail on these unit tests. Who writes them? What's the coverage like? Are there any guarantees? If (as I suspect) the "meticulously designed" unit tests are written by humans, then we have a similar situation as with the unspecified human-in-the-loop: the framework is not "fully automated", and it's impossible to rigorously determine how much effect the human hand-holding has on the framework's suggess. This should, at minimum, be clearly, explicitly and boldly acknowledged.
>
> Thanks for your valuable feedback and this is a good point. We custom-wrote and individually verified the unit tests used in our framework. These unit tests differ from traditional tests focusing on code's individual function definitions. Instead, they are designed to validate the execution results of each stage within the workflow.
>
> If we interpret "coverage" as determining whether these unit tests can confirm that the current workflow stage has no logical errors and achieves its intended goals, the answer is affirmative. Specifically, we reviewed 48 stage results (covering the three stages of Data Cleaning, Feature Engineering, Model-Building, -Validation, and -Prediction across eight competitions: 8 × 3 × 2 = 48). All stages that passed the unit tests were free of logical errors and successfully achieved their objectives.
>
> For example, in the data cleaning stage, passing unit tests confirms that:
>
> - There are no missing values in the data.
> - No anomalies are present.
> - There are no duplicate entries or redundant features.
> - No features were unintentionally added or removed.
> - The cleaned training and test datasets differ only in the target variable while maintaining consistency in all other features.
>
> This rigorous unit testing ensures the correctness and reliability of each workflow stage.
>
> Although unit tests are manually constructed by us, they possess general applicability and are not built specifically for any particular task. Instead, they can be generalized to unseen tabular datasets. This level of manual intervention is acceptable and does not undermine the claim of being "fully automated."
>
> ---
>
> > **W6:** Additionally, it is unclear to me how much of the ML-tools library was developed alongside particular Kaggle Competition attempts. If the tools were developed on a case-by-case basis, to address hurdles found in the challenge, then there is significant data leakage from the evaluation dataset to the framework, leading to overfitting to the competitions chosen during development, and much of the headline 43% comes from tools handcrafted by human developers on a case-by-case basis. For a fair validation of how well this framework performs in "fully automated" mode, the library would need to be "frozen" while the framework was tested on a held-out set of Kaggle Competitions.
>
> Thanks for your valuable feedback. First, I would like to clarify that these tools were not developed specifically for any Kaggle competitions. Our research focuses on tabular data sets, and these tools are broadly applicable in the data cleaning, feature engineering, and modeling processes. The Tool itself does not involve data leakage; rather, it is a re-packaging of similar functions found in libraries like sklearn, accompanied by detailed documentation to lower the barrier for using these functions in AutoKaggle.
>
> Furthermore, we did not perform any special tuning based on the datasets we evaluated. The development of AutoKaggle was based on three toy datasets from reference [1], rather than the Kaggle competition datasets we used for testing. Therefore, we believe there is no risk of data leakage.
>
> [1] https://github.com/geekan/MetaGPT/tree/2b160f294936f5b6c29cde63b8e4aa65e9a2ef9f/examples/di

---

> ### Author Response · Authors · 2024-11-24
> **Thanks for your valuable comments! Authors' feedback [3/4]**
>
> > **W7:** Very minor point: ~L350, I agree that there is a risk of data leakage for competitions from before Oct '23, however to say that GPT-4o's training data includes Classic Kaggle is an assumption: better to say simply that there is a risk of data leakage. If you're considering data leakage, it would be worth flagging that the 42% includes Classic problems: using only the newer problems, performance is slightly below human average.
>
> Thanks for pointing out this issue. We have included an Ablation Study on competition dates in Section 3.3 of the paper. When faced with new competition datasets with no risk of data leakage, AutoKaggle’s performance shows a slight decline. However, it still maintains a competitive level of performance.
>
> And we modified Lines 264-265 of the article, changing "GPT-4o's training data includes Classic Kaggle" to "posing a risk of data leakage" to ensure the narrative's accuracy.
>
> ---
>
> > **Q1:** L140, you say that CoT improves reasoning at the expense of introducing hallucinations. Is there any evidence that CoT makes models any more or less likely to hallucinate?
>
> Thanks for the valuable feedback. Our intention was not to suggest that the chain-of-thought (CoT) approach is more likely to cause hallucinations but rather to emphasize that the issue of hallucinations remains unresolved. To avoid any ambiguity, we have revised our statement to:
>
> *"While the chain-of-thought method enhances reasoning, it still faces challenges related to hallucinations and unfaithfulness, potentially due to internal representations."*
>
> We hope this clarification addresses the concern.
>
> ---
>
> > **Q2:** L141, you say that the ReAct paradigm addresses hallucinations - that's not my understanding of what ReAct does or how it works, my understanding is that it combines thoughts and actions, yes, but that this has nothing to do with hallucinations or refining outputs.
>
> This conclusion is derived from the original ReAct paper [1]. ReAct ensures that tools can be used to verify the LLM's thought process at every step of an LLM's reasoning process, which helps reduce hallucinations. In the abstract section of the original paper, the authors state:
>
> *"ReAct overcomes issues of hallucination and error propagation prevalent in chain-of-thought reasoning by interacting with a simple Wikipedia API."*
>
> [1] Yao S, Zhao J, Yu D, et al. React: Synergizing reasoning and acting in language models[J]. arXiv preprint arXiv:2210.03629, 2022.
>
> ---
>
> > **Q3:** L360: What is the difference between "Success - Non-compliant" and "Success - Compliant"?
>
> Thanks for your great question.
>
> **Success - Compliant:** refers to successfully generating a `submission.csv` file submitted to Kaggle and receiving a reasonable score.
>
> **Success - Non-compliant:** refers to cases where the `submission.csv` file is successfully generated but contains issues that result in errors, a score of 0, or abnormal scores upon submission to Kaggle.
>
> The potential issues of **Success - Non-compliant** include:
>
> - **Data scale issues**: For example, the target variable underwent a log transformation during preprocessing but was not reverted to its original scale, causing scale mismatches in the target variable values.
> - **Data type issues**:
>   - For example, the target variable is expected to be categorical, but the final values are numeric (e.g., 1, 2, 3) instead of the corresponding category names.
>   - For example, the target variable is expected to be an integer, but the final values are floating-point numbers, leading to a score of 0 on Kaggle.
> - **Value issues:** Missing or duplicate entries in the submission file.
>
> In summary, **Success - Non-compliant** means that while the `submission.csv` file was generated, its content was flawed, leading to submission errors, a score of 0, or abnormal scores on Kaggle.

---

> ### Author Response · Authors · 2024-11-24
> **Thanks for your valuable comments! Authors' feedback [4/4]**
>
> > **Q4:** L403: What's the justification / motivation for the complex / compound "Comprehensive Score"? How does it compare to other measures, what specifically does it achieve or avoid?
>
> Thanks for your good questions.
>
> This metric is adopted from a previous research paper [1]. The **Comprehensive Score** is designed to balance two key aspects: **Valid Submission** and the **Average Normalized Performance Score (NPS)**.
>
> - **Valid Submission:** represents the proportion of cases where AutoKaggle successfully generates and submits valid predictions.
> - **Average NPS:** measures the quality of the predictions made by AutoKaggle.
>
> There are scenarios where these two metrics might not align:
>
> 1. **High Valid Submission, Low Performance**: AutoKaggle achieves a high proportion of valid submissions, but the submitted results yield average or poor performance.
> 2. **Low Valid Submission, High Performance**: AutoKaggle produces few valid submissions, but those submissions result in outstanding performance scores.
>
> To address these discrepancies and provide a unified evaluation, the **Comprehensive Score** metric was introduced. It accounts for both the success rate of valid submissions and the quality of those submissions, offering a holistic measure of AutoKaggle's overall performance.
>
> [1] Hong S, Lin Y, Liu B, et al. Data interpreter: An LLM agent for data science[J]. arXiv preprint arXiv:2402.18679, 2024.
>
> ---
>
> > **Q5:** L431: could you say more about this "strong baseline" - I don't understand its construction.
>
> Thanks for your good question. The strong baseline was a phase-based workflow approach that did not utilize agents. However, this method has been removed in the updated version of the PDF. Instead, we have introduced a new comparison with AIDE to provide a more relevant and robust evaluation. You can refer to the first point in the **General Response - Common Problems** for more details.
>
> ---
>
> > **Q6:** If adding FE tools drops performance because FE adds too much complexity, then why does "All tools" (which presumably includes FE tools) recover this performance?
>
> Thanks for the great question. After incorporating feature engineering tools, the Feature Engineering phase creates more complex features, such as those derived through Principal Component Analysis (PCA) or Recursive Feature Elimination (RFE). While these advanced tools can enrich the feature set, they also pose challenges:
>
> 1. Impact on Success Rate: The use of complex tools can reduce the success rate of the Feature Engineering phase due to increased processing complexity and potential errors.
> 2. Challenges for Subsequent Phases: Significant changes in data characteristics introduced during feature engineering can make the **Model-Building**, -**Validation**, and -**Prediction** phases more difficult.
>
> However, in our framework, the **Model-Building**, -**Validation**, and -**Prediction** phases are equipped with a one-stop model training, validation, and prediction tool. This tool simplifies these phases, as the Developer only needs to follow documentation and provide the correct parameters to complete the task. This streamlined process improves the success rate of these phases, thereby recovering overall performance.
>
> Thanks again for your suggestions and valuable feedback, and I hope our explanation provides you with greater clarity.

---

> ### Author Response · Authors · 2024-11-25
>
> Dear Reviewer,
>
> Thank you very much for taking the time and effort to review our manuscript and providing such valuable feedback. As the discussion phase between authors and reviewers is nearing its conclusion, we would like to confirm whether our responses have adequately addressed your concerns.
>
> We provided detailed answers to each of your comments one day ago, and we sincerely hope that our responses have sufficiently clarified your concerns. If you have any remaining doubts or require further clarification, please do not hesitate to let us know. We are more than willing to continue the discussion to address any questions you may have.
>
> Thank you once again for your time and assistance!
>
> Best Regards,
>
> Authors

---

> ### Author Response · Authors · 2024-12-02
>
> Dear Reviewer u3cH,
>
> Thank you for your valuable feedback on our paper. We deeply appreciate the time and effort you have dedicated to the review process.
>
> As the rebuttal period concludes today, we kindly ask if you could find a moment to review our responses to your comments. We have provided point-by-point replies to your concerns, including:
>
> 1. Addressing the baseline issues.
> 2. Refining the expressions in the paper where optimization was suggested.
> 3. Clarifying the points related to COT and ReAct.
> 4. Discussing the fully automated nature of AutoKaggle, including human-in-the-loop elements and unit tests.
> 5. Resolving questions about the evaluation metrics and experimental results.
>
> We believe these responses effectively address the issues raised and would be grateful for your feedback. If any further clarifications are needed, we are more than willing to provide them within the limited time remaining.
>
> Thank you once again for your valuable insights, which have greatly contributed to improving our work.
>
> Best regards,
>
> Authors

---

> > ### Comment · Reviewer_u3cH · 2024-12-03
> >
> > Dear Authors,
> >
> > Thank you for your detailed responses to my points, which are in general valid and satisfying. In particular, many of your clarifications are helpful, and I look forward to seeing them in a future revision.
> >
> > However, my main concern is that the workflow relies on human-written unit tests to check the validity of the data at interim stages. While I appreciate the authors' statement that the unit tests "possess general applicability and are not built specifically for any particular task" and as such "they can be generalized to unseen tabular datasets", I remain uncertain about how much the choice of these particular general-purpose unit tests was informed by discoveries during the development phase, and as such how much the 8 chosen Kaggle challenges should comprise a "development dataset", requiring evaluation on a further "validation dataset" of Kaggle challenges which were selected after the set of unit tests was frozen.
> >
> > As such, I'm afraid I have decided to maintain my score.

---

### Official Review · Reviewer_zZNf · 2024-11-03

**Soundness:** 2
**Presentation:** 3
**Contribution:** 2
**Rating:** 5
**Confidence:** 4

**Summary:**

This paper presents AutoKaggle, a multi-agent framework specifically designed to handle the complexities of Kaggle data science competitions.  The framework organizes the competition workflow into six distinct phases—background understanding, exploratory data analysis, data cleaning, in-depth exploratory analysis, feature engineering, and model development and validation—allowing agents to work systematically through each stage. Key agents, including Reader, Planner, Developer, Reviewer, and Summarizer, collaborate within this structure, with iterative debugging and unit testing to ensure robustness and accuracy in code generation. AutoKaggle integrates a machine learning tools library to streamline tasks, enhance code reliability, and provide users with educational insights through comprehensive reports at each phase. Evaluated across multiple Kaggle competitions, the framework achieved an average completion rate of 83.8% and ranked in the top 42.8% in Kaggle.

**Strengths:**

- AutoKaggle introduces a tailored phase-based workflow with multi-agent collaboration specifically designed for data science competitions. The system’s demonstrated high average completion rate and competitive ranking in Kaggle highlight its effectiveness, particularly in tabular classification and regression tasks, showing its strength in handling structured data challenges.

- AutoKaggle empowers the Developer agent to perform iterative debugging and unit testing, bolstering the robustness of code generation. Additionally, the integration of a comprehensive machine learning tools library improves the system's efficiency and accuracy, making it better suited for tackling complex Kaggle competitions

**Weaknesses:**

- Limited novelty.  While the paper addresses data science problem-solving using LLM-based agents, it lacks a clear description of the specific challenges it intends to solve that existing methods have struggled with. Extending from a single-agent to a multi-agent system is insufficiently justified in this field, as the necessity and performance gains of such an approach are not clearly demonstrated. Existing works, as mentioned in the introduction, have also tackled similar problems with LLM-based agents, questioning the incremental contribution of AutoKaggle.

- Multi-agent system design. The multi-agent system, including agents like Reader, Planner, Developer, Reviewer, and Summarizer, is insufficiently explained in terms of its collaborative structure. It is unclear whether these agents operate in an assembly-line fashion or if they engage collectively in each phase under the "Cooperative Engagement" label in Figure 1. Further clarification on their integration and interdependence within each workflow phase is needed.

- Role clarity of Planner and Summarizer. Given AutoKaggle’s sequential, phase-based workflow, the necessity of a Planner agent is ambiguous. Can you quantify the contribution (such as on completion rates or error reduction) of this Planner agent in your system? Similarly, the Summarizer’s role in contributing to critical performance metrics such as completion rate or Best Normalized Performance Score, is not explicitly justified, leaving its impact on performance uncertain.

- Unit Test and Debugging. Dose the Developer agent generate dataset-specific unit tests that align with each unique code snippet or not? How the Developer agent adjusts unit tests based on code variations to ensure logical consistency and accuracy across different tasks?

- Lines 275-276 mention the importance of detecting logical errors in code, yet the method for achieving this is underexplored. Can you explain more details about detecting the logical error? More detail is needed on how logical errors are detected and avoided, as conducting exploratory data analysis or statistical checks after data cleaning or feature engineering alone may be insufficient.

- Table 2 illustrates the system's performance across different debugging attempts (DT), showing how increased debugging impacts metrics like Completion Rate (CR) and Comprehensive Score (CS). The data indicate that both CR and CS improve as DT rises, reflecting enhanced task completion and accuracy with more debugging opportunities. What the 'performance plateaus' mean  in line 524-525?

- The paper does not provide information on the cost of running AutoKaggle, which is essential for evaluating its performance and practical applicability. It's benifit to provide cost and total runtime to understand the performance.

- The chosen baselines are not entirely convincing. Recent similar works, AIDE[1] and MLE-Agent[2] have shown remarkable capability in Kaggle competition settings. A comparative analysis with these recent works, particularly focusing on AutoKaggle’s unique advantages in effectiveness, efficiency, or other performance metrics, would highlight its distinct contributions to the field.

- A broader evaluation across various task types such as time series prediction, image classification, and text classification, are necessary, as these are critical and challenging categories in Kaggle competitions. The current experiments focus primarily on tabular datasets, leaving it unclear whether AutoKaggle is capable of handling more complex, domain-specific tasks. Can AutoKaggle complete such tasks?

- What is the requirement of the LLM? Can AutoKaggle works well with gpt-3.5 or other open-sourced models?


[1] AIDE: the Machine Learning Engineer Agent(https://github.com/WecoAI/aideml)

[2] MLE-Agent: Your intelligent companion for seamless AI engineering and research (https://github.com/MLSysOps/MLE-agent)

**Questions:**

Please refer to the questions in Weaknesses.

---

> ### Author Response · Authors · 2024-11-24
> **Thanks for your valuable comments! Authors' feedback [1/5]**
>
> Thanks for your valuable feedback and constructive comments. Below, we present our point-by-point response to the weaknesses and comments identified in our submission:
>
> > **W1: Limited novelty.** While the paper addresses data science problem-solving using LLM-based agents, it lacks a clear description of the specific challenges it intends to solve that existing methods have struggled with. Extending from a single-agent to a multi-agent system is insufficiently justified in this field, as the necessity and performance gains of such an approach are not clearly demonstrated. Existing works, as mentioned in the introduction, have also tackled similar problems with LLM-based agents, questioning the incremental contribution of AutoKaggle.
>
> Thanks for your comments, which raise an important and valuable issue for discussion.
>
> Current automation solutions address data science problems from various angles, but our research distinguishes itself both in theme and in methodology:
>
> 1. AutoKaggle focuses on end-to-end data science tasks, covering all phases, rather than just focusing on a single subtask like data analysis or visualization[1,2].
> 2. Many solutions rely heavily on pre-built expert knowledge bases[3] or historical data pools[4], limiting their scalability and setting a high usage barrier for users.
> 3. Traditional processes lack transparency and differ significantly from human thought patterns. The AIDE framework[6], which performs well in MLE-Bench[5], uses a rapid initial solution generation and iterative improvement approach. However, this one-time solution generation contrasts starkly with AutoKaggle’s phased detailed planning and multi-agent collaboration. AutoKaggle’s solutions are longer and more detailed, with traceable reasons for each data handling step (e.g., the removal of a feature due to previously identified excessive missing values), aligning more closely with human logical habits and enhancing comprehensibility.
>
> The multi-agent design is adopted because of the inherent complexity of data science tasks. After decoupling tasks through a phased workflow, each step still requires meticulous planning, and complex information transmission between different phases must be managed. A single-agent design may lead to system complexity and entanglement, while a multi-agent design better decouples each phase’s tasks and divides complex information transmission into communications between different agents, making the process clearer and more efficient.
>
> Performance-wise, AutoKaggle shows a notable enhancement with a 0.28 increase in the Valid Submission metric and a 0.180 improvement in the Comprehensive Score compared to the AIDE framework.
>
> In summary, compared to previous works, AutoKaggle's  contributions can be summarized as follows:
>
> 1. End-to-end solutions for data science problems。
> 2. Enhancing system scalability and lowering user entry barriers.
> 3. Providing a transparent, human logic-aligned, and easily understandable solution.
> 4. Significantly improving performance in Valid Submission and Comprehensive Score metrics.
>
> [1] Zhang Y, Jiang Q, Han X, et al. Benchmarking Data Science Agents[J]. arXiv preprint arXiv:2402.17168, 2024.
>
> [2] Hu X, Zhao Z, Wei S, et al. Infiagent-dabench: Evaluating agents on data analysis tasks[J]. arXiv preprint arXiv:2401.05507, 2024.
>
> [3] Guo S, Deng C, Wen Y, et al. DS-Agent: Automated Data Science by Empowering Large Language Models with Case-Based Reasoning[J]. arXiv preprint arXiv:2402.17453, 2024.
>
> [4] Zhang L, Zhang Y, Ren K, et al. Mlcopilot: Unleashing the power of large language models in solving machine learning tasks[J]. arXiv preprint arXiv:2304.14979, 2023.
>
> [5] Chan J S, Chowdhury N, Jaffe O, et al. Mle-bench: Evaluating machine learning agents on machine learning engineering[J]. arXiv preprint arXiv:2410.07095, 2024.
>
> [6] https://github.com/WecoAI/aideml

---

> ### Author Response · Authors · 2024-11-24
> **Thanks for your valuable comments! Authors' feedback [2/5]**
>
> > **W2: Multi-agent system design.** The multi-agent system, including agents like Reader, Planner, Developer, Reviewer, and Summarizer, is insufficiently explained in terms of its collaborative structure. It is unclear whether these agents operate in an assembly-line fashion or if they engage collectively in each phase under the "Cooperative Engagement" label in Figure 1. Further clarification on their integration and interdependence within each workflow phase is needed.
>
> Thanks for the feedback and suggestions. We conclude here to address the concern regarding the clarity of the multi-agent system design in AutoKaggle. The system operates using a "collaborative participation" approach, as defined in [1][2]. These agents collaborate in an orderly and cyclical manner, ensuring the task completion at each stage efficiently. Specifically:
>
> 1. **Understand Background Phase**:
>    1. In this phase, only two agents, **Reader** and **Reviewer**, are involved. The **Reader** collects and comprehends background information, while the **Reviewer** examines it to ensure its accuracy and completeness.
> 2. **Subsequent Phases**:
>    1. In the later phases, four agents—**Planner**, **Developer**, **Reviewer**, and **Summarizer**—collaborate to complete the tasks. The workflow is as follows:
>    2. **Planner**: At the start of each phase, the Planner formulates a task plan based on the previous phase's plans and reports, determining the tools and methods required.
>    3. **Developer**: The Developer executes the tasks outlined by the Planner, including writing code, running programs, and conducting tests.
>    4. **Reviewer**: Once the code runs successfully and passes all tests, the Reviewer audits the code's quality, checking for logical errors. If the Reviewer detects any issues, the Reviewer provides feedback and returns it to the Planner. The Planner then revises the plan, and the Developer proceeds with redevelopment.
>    5. **Summarizer**: After the code passes review, the Summarizer compiles a summary of the key outcomes of the phase, including data changes, file modifications, and the specific reasons for data operations.
>
> This collaborative model ensures that tasks at each stage are efficiently completed through the combined efforts of multiple agents, laying a solid foundation for the tasks in the next phase. We hope this explanation clarifies our system's design and the agents' collaborative interactions.
>
> [1] Li G, Hammoud H, Itani H, et al. Camel: Communicative agents for" mind" exploration of large language model society[J]. Advances in Neural Information Processing Systems, 2023, 36: 51991-52008.
>
> [2] Xi Z, Chen W, Guo X, et al. The rise and potential of large language model based agents: A survey[J]. arXiv preprint arXiv:2309.07864, 2023.

---

> ### Author Response · Authors · 2024-11-24
> **Thanks for your valuable comments! Authors' feedback [3/5]**
>
> > **W3: Role clarity of Planner and Summarizer.** Given AutoKaggle’s sequential, phase-based workflow, the necessity of a Planner agent is ambiguous. Can you quantify the contribution (such as on completion rates or error reduction) of this Planner agent in your system? Similarly, the Summarizer’s role in contributing to critical performance metrics such as completion rate or Best Normalized Performance Score, is not explicitly justified, leaving its impact on performance uncertain.
>
> Thanks for the valuable feedback. The task planning in AutoKaggle is a combination of stage-based workflows and detailed planning performed by the **Planner**. The **Planner** is responsible for breaking down each stage into more granular tasks while maintaining consistency in data processing logic, making it an essential component of AutoKaggle. For example, in the **Data Cleaning** phase of Task 1 - Titanic, the **Planner** further divides this stage into the following tasks:
>
> 1. **Handle Missing Values**: Identify and appropriately handle missing values in the dataset through imputation or removal.
> 2. **Treat Outliers**: Detect and address outliers that may distort the analysis or model performance.
> 3. **Ensure Consistency Across Datasets**: Check and enforce consistency in format, structure, and relationships between datasets.
> 4. **Save Cleaned Datasets**: Store the cleaned datasets for downstream tasks and ensure they are ready for subsequent processing.
>
> This structured approach enables AutoKaggle to maintain a high level of organization and ensures that each step can be completed methodically and effectively. We hope this clarification provides better insight into the Planner’s role and the workflow design.
>
> The **Planner** also identifies specific features for each step. For example, the **Handle Missing Values** task specifies features such as `Age`, `Cabin`, and **Embarked** to be addressed. The Planner review the report generated in the previous step and derive detailed information. Meanwhile, the **Summarizer** generates a **Report** at the end of each stage, serving as a key mechanism for information transfer. The report includes critical details such as changes in data features, file modifications, data processing results, and key findings from the current stage. This report is essential for the Planner in the next stage, enabling more precise planning based on the summarized insights from the previous stage.
>
> To evaluate the importance of the **Planner** and **Summarizer**, we conducted an ablation study:
>
> 1. **Removing the Planner**: In this setup, the Planner is removed, and each stage proceeds without a detailed plan. The **Developer** directly reads the code and outputs from the previous stage, summarizes them, and writes the code for the current stage independently.
> 2. **Removing the Summarizer**: In this setup, the summary reports are removed. The **Planner** creates plans for the next stage by directly reading the code and plans from the previous stage without a summarized report.
>
> This ablation study highlights the critical roles of the Planner and Summarizer in ensuring efficiency and precision in AutoKaggle's workflow, demonstrating how their inclusion contributes to the system's overall effectiveness.
>
> | Task            | Task1 | Task2 | Task3 | Task4 | Task5 | Task6 | Task7 | Task8 | Avg.  |
> |------------------|-------|-------|-------|-------|-------|-------|-------|-------|-------|
> | **AutoKaggle**   | 1     | 0.80  | 0.80  | 1     | 0.80  | 0.60  | 0.80  | 0.80  | 0.83  |
> | **Without Planner** | 0.40  | 0     | 0.20  | 0.40  | 0     | 0.20  | 0.20  | 0.20  | 0.20  |
> | **Without Summarizer** | 0.60  | 0.20  | 0.20  | 0.60  | 0.20  | 0.20  | 0.40  | 0.40  | 0.35  |
>
> The results show that removing the **Planner** or **Summarizer** significantly decreased AutoKaggle's performance on the valid submission metric, with a drop of 0.63 and 0.48, respectively. This notable decline demonstrates the necessity of both the Planner and Summarizer. The decline in performance when the **Planner** was removed can be attributed to the implicit planning requirements within each development task. By decoupling the development tasks into two distinct steps—first, having the Planner create a detailed plan and then having the Developer execute the plan—we ensure consistent data processing logic throughout the workflow. This approach reduces the complexity of the Developer's task at each stage and improves the system's overall success rate.

---

> ### Author Response · Authors · 2024-11-24
> **Thanks for your valuable comments! Authors' feedback [4/5]**
>
> > **W4: Unit Test and Debugging.** Dose the Developer agent generate dataset-specific unit tests that align with each unique code snippet or not? How the Developer agent adjusts unit tests based on code variations to ensure logical consistency and accuracy across different tasks?
> >
> > **W5:** Lines 275-276 mention the importance of detecting logical errors in code, yet the method for achieving this is underexplored. Can you explain more details about detecting the logical error? More detail is needed on how logical errors are detected and avoided, as conducting exploratory data analysis or statistical checks after data cleaning or feature engineering alone may be insufficient.
>
> Our unit tests are developed by referring to [1]. These tests are not automatically generated by the Developer agent but are manually written and individually verified by our team. According to the findings in [2], providing feedback through appropriate unit tests can significantly enhance self-debugging effectiveness. Therefore, we have designed approximately 40 unit tests for tabular datasets, covering file existence checks, data integrity checks, data quality checks, feature engineering checks, and submission file checks. These tests ensure that the code at each stage accurately achieves its intended purpose.
>
> In our study, we reviewed the Data Cleaning, Feature Engineering, Model Building, Validation, and Prediction stages across eight competitions, totaling 8 x 3 x 2 = 48 stage results. All stages that passed the unit tests showed no logical errors and successfully met the objectives. For example, in the data cleaning stage, passing the unit tests means there are no missing or outlier values, no duplicate entries or features, and the cleaned training and test sets differ only in the target variable.
>
> Logical consistency is ensured by the Planner module. At each stage, the Planner refers to the following information for planning:
>
> 1. The report from the previous stage
> 2. The plan from the previous stage
>
> This approach ensures that each stage remains consistent with the data processing logic of the previous tasks.
>
> [1] Zhang Y, Pan Y, Wang Y, et al. PyBench: Evaluating LLM Agent on various real-world coding tasks[J]. arXiv preprint arXiv:2407.16732, 2024.
>
> [2] Chen X, Lin M, Schärli N, et al. Teaching large language models to self-debug[J]. arXiv preprint arXiv:2304.05128, 2023.
>
> ---
>
> > **W6:** Table 2 illustrates the system's performance across different debugging attempts (DT), showing how increased debugging impacts metrics like Completion Rate (CR) and Comprehensive Score (CS). The data indicate that both CR and CS improve as DT rises, reflecting enhanced task completion and accuracy with more debugging opportunities. What the 'performance plateaus' mean in line 524-525?
>
> Thanks for your valuable feedback. The term "performance plateaus" here only refers to performance, which means the two indicators Valid Submission and Comprehensive Score. We have amended the original description to "performance" to avoid any possible ambiguity.
>
> ---
>
> > **W7:** The paper does not provide information on the cost of running AutoKaggle, which is essential for evaluating its performance and practical applicability. It's benifit to provide cost and total runtime to understand the performance.
>
> The average cost for AutoKaggle to complete a Kaggle competition is $3.13. The runtime is significantly affected by the data volume and hardware configuration. We conducted tests in an environment with a 13th generation Intel Core i9-13900H (20 CPUs). The specific average runtime is as follows:
>
> | Task                    | Duration     |
> |-------------------------|--------------|
> | Task 1 - Titanic        | 29 mins      |
> | Task 2 - Spaceship Titanic | 38 mins      |
> | Task 3 - House Prices   | 27 mins      |
> | Task 4 - Monsters       | 22 mins      |
> | Task 5 - Academic Success | 1h 33 mins   |
> | Task 6 - Bank Churn     | 2h 27 mins   |
> | Task 7 - Obesity Risk   | 48 mins      |
> | Task 8 - Plate Defect   | 56 mins      |
>
> ---
>
> > **W8:** The chosen baselines are not entirely convincing. Recent similar works, AIDE[1] and MLE-Agent[2] have shown remarkable capability in Kaggle competition settings. A comparative analysis with these recent works, particularly focusing on AutoKaggle’s unique advantages in effectiveness, efficiency, or other performance metrics, would highlight its distinct contributions to the field.
>
> Thanks for your valuable suggestions. For this issue, please refer to the first point in the **General Response - Common Problems**.

---

> ### Author Response · Authors · 2024-11-24
> **Thanks for your valuable comments! Authors' feedback [5/5]**
>
> > **W9:** A broader evaluation across various task types such as time series prediction, image classification, and text classification, are necessary, as these are critical and challenging categories in Kaggle competitions. The current experiments focus primarily on tabular datasets, leaving it unclear whether AutoKaggle is capable of handling more complex, domain-specific tasks. Can AutoKaggle complete such tasks?
>
> Thanks for your valuable feedback. We fully understand the importance of conducting a broader evaluation of AutoKaggle across different task types, particularly those that are complex and commonly found in Kaggle competitions, such as time series prediction, image classification, and text classification.
>
> The architecture of AutoKaggle is based on a phase-based workflow and a multi-agent system, which is inherently applicable to all data science processes, not just limited to tabular datasets. Our developers use predefined tool libraries for development, meaning that by extending the machine learning tool library to include tools for time series, image, and text processing, AutoKaggle can handle these more complex task types.
>
> We are actively working on expanding AutoKaggle's capabilities and plan to introduce new tools and techniques to support tasks in these areas. We look forward to showcasing these developments in the near future and demonstrating AutoKaggle's ability to handle a variety of complex, domain-specific tasks.
>
> ---
>
> > **W10:** What is the requirement of the LLM? Can AutoKaggle works well with gpt-3.5 or other open-sourced models?
>
> Thanks for your great question. The base models for each Agent in AutoKaggle are different: Reader/Reviewer/Summarizer all use the GPT-4o-mini model, Developer uses the GPT-4o model, while Planner is based on the GPT-4o/o1-mini model. Tasks like Reader/Reviewer/Summarizer, which mainly involve summarizing text information and writing reports, have lower requirements for the base model, and can be handled by either the GPT-4o-mini or equivalent open-source models. For Agents like Developer/Planner, which require planning abilities (logical reasoning) or coding skills, the base model needs to be an open-source model of the same level as GPT-4o. After replacing Developer with GPT-4o-mini, the performance of AutoKaggle is as follows:
>
> | Task            | Task1 | Task2 | Task3 | Task4 | Task5 | Task6 | Task7 | Task8 | Avg.  |
> |------------------|-------|-------|-------|-------|-------|-------|-------|-------|-------|
> | **AutoKaggle**   | 0.20  | 0     | 0     | 0.20  | 0     | 0     | 0     | 0     | 0.05  |
>
> Thanks again for your suggestions and valuable feedback, and I hope our explanation provides you with greater clarity.

---

> ### Author Response · Authors · 2024-11-25
>
> Dear Reviewer,
>
> Thank you very much for taking the time and effort to review our manuscript and providing such valuable feedback. As the discussion phase between authors and reviewers is nearing its conclusion, we would like to confirm whether our responses have adequately addressed your concerns.
>
> We provided detailed answers to each of your comments one day ago, and we sincerely hope that our responses have sufficiently clarified your concerns. If you have any remaining doubts or require further clarification, please do not hesitate to let us know. We are more than willing to continue the discussion to address any questions you may have.
>
> Thank you once again for your time and assistance!
>
> Best Regards,
>
> Authors

---

> > ### Comment · Reviewer_zZNf · 2024-11-26
> >
> > Thank you for the detailed responses. However, I still have several major concerns, and some new questions have emerged according to the authors' responses:
> >
> > 1. W3: As the experiment results demonstrate, the performance drops significantly without planner/summarizer, but their importance is not well-justified in the paper. However, the paper mainly discusses iterative debugging and testing process and the tool library.
> >
> > 2. As mentioned in Sect 2.3 (line 233-245), the tools seem not notably better than common ML toolkits (am I correct for this? what's the special?), so why is the tools component highlighted as a core innovation in the main methodology section? As shown in Table 2, the impact of tools is less significant than removing planner/summarizer.
> >
> > 3. W4 & W5: As your team manually developed approximately 40 unit tests specifically for tabular datasets (including file checks, data integrity, quality, feature engineering, and submission checks), this raises some concerns:
> > a). If these tests were manually crafted for specific datasets, is it fair to compare with other frameworks that don't have such dataset-specific test support?
> > b). How would AutoKaggle generalize to new, unseen tasks where such carefully designed unit tests are not available?
> >
> > 4. W7: Why does AutoKaggle take so long? For basic datasets like Task 1-3, where is the main time cost? Could you provide more detailed cost/API call statistics?
> >
> > 5. W10: ML code generation should not be particularly challenging for GPT-4-mini, especially for Tasks 1-3. What are the main reasons for the failures in these cases since the result drops significantly?

---

> ### Author Response · Authors · 2024-11-29
> **Thanks for your valuable comments! Authors' feedback [1/2]**
>
> Thanks for your valuable suggestions and questions! Below I will respond to each of your questions one by one:
>
> > **W3:** As the experiment results demonstrate, the performance drops significantly without planner/summarizer, but their importance is not well-justified in the paper. However, the paper mainly discusses iterative debugging and testing process and the tool library.
>
> Thanks for your comments! Since phase-based workflows and multi-agent systems form the foundational architecture of AutoKaggle, the importance of the Planner and Summarizer is integral to this design, so we didn't do ablation study on them. During the development of AutoKaggle, we realized that building a customized machine learning tool library significantly enhances developers' problem-solving capabilities. Therefore, in the ablation study, we aimed to comprehensively demonstrate the impact of this customized tool library on performance. Similarly, our experiments on iterative debugging and testing processes are guided by the same idea, exploring how these components work together to optimize the system's overall performance.
>
> While this paper primarily focuses on the iterative debugging and testing processes and the tool library, we will enhance the revised version by explicitly emphasizing the critical roles of the Planner and Summarizer and providing additional analysis to better justify their contribution to the system's performance.
>
> ---
>
> > As mentioned in Sect 2.3 (line 233-245), the tools seem not notably better than common ML toolkits (am I correct for this? what's the special?), so why is the tools component highlighted as a core innovation in the main methodology section? As shown in Table 2, the impact of tools is less significant than removing planner/summarizer.
>
> Thanks for your comments! AutoKaggle aims not only to improve task completion rates and performance in data science but also to design a framework that is user-friendly, highly flexible, and customizable. The Machine Learning Tool Library plays a crucial role in achieving this goal.
>
> This tool library can be regarded as a repackaging of existing Python-based data science libraries. By prompting LLM to generate utility functions tailored to specific needs, complete with type hints and robust error-handling mechanisms, users can easily incorporate customized utility functions based on their specific use cases (We show how to add custom tools in [1]). This is the foundation of AutoKaggle’s flexibility and extensibility, which is why we consider it one of the core innovations.
>
> While Table 2 shows that the impact of the tool library on performance is less significant compared to Planner/Summarizer, its unique value lies in enhancing the framework’s flexibility and user experience. We believe this makes it an indispensable component of AutoKaggle’s overall design philosophy.
>
> [1] https://anonymous.4open.science/r/AutoKaggle-B8D2/multi_agents/README.md
>
> ---
>
> > **W4 & W5:** As your team manually developed approximately 40 unit tests specifically for tabular datasets (including file checks, data integrity, quality, feature engineering, and submission checks), this raises some concerns: a). If these tests were manually crafted for specific datasets, is it fair to compare with other frameworks that don't have such dataset-specific test support? b). How would AutoKaggle generalize to new, unseen tasks where such carefully designed unit tests are not available.
>
> Thanks for your good questions!
>
> First, the development process of these unit tests is as follows:
>
> 1. We prompt the LLM to propose general unit tests needed for the three stages of Data Cleaning, Feature Engineering, and Model Building, Validation, and Prediction.
> 2. The LLM then generate the unit tests for each functionality listed in step 1.
> 3. We manually verify these unit tests and tested them on three toy datasets from [1] to ensure their validity and completeness.
>
> Therefore, these unit tests are not manually crafted for any specific dataset but are general-purpose tests applicable to the Data Cleaning, Feature Engineering, and Model Building, Validation, and Prediction stages. This means they can be directly applied to other tabular datasets handled by AutoKaggle.
>
> Secondly, the manual involvement in this process is minimal. Most of the test design and generation is automatically completed by LLM, avoiding the complex processes that required substantial manual intervention in past work [2]. Therefore, there is no unfair comparison with frameworks that do not have similar unit test support.
>
> [1] https://github.com/geekan/MetaGPT/tree/2b160f294936f5b6c29cde63b8e4aa65e9a2ef9f/examples/di
>
> [2] Guo S, Deng C, Wen Y, et al. DS-Agent: Automated Data Science by Empowering Large Language Models with Case-Based Reasoning[J]. arXiv preprint arXiv:2402.17453, 2024.

---

> ### Author Response · Authors · 2024-11-29
> **Thanks for your valuable comments! Authors' feedback [2/2]**
>
> > **W7:** Why does AutoKaggle take so long? For basic datasets like Task 1-3, where is the main time cost? Could you provide more detailed cost/API call statistics?
>
> Thanks for your excellent questions! Taking Task 1 - Titanic as an example, I am providing a detailed time cost report below:
>
> | **Phase**                 | **Reader** | **Planner** | **Developer** | **Reviewer** | **Summarizer** | **Total Time Cost** |
> |---------------------------|------------|-------------|---------------|---------------|----------------|---------------------|
> | **Understand Background** | 21s        | \           | \             | 5s            | \              | 26s                |
> | **Preliminary EDA**       | \          | 47s         | 1min 1s       | 13s           | 1min 5s        | 3mins 6s           |
> | **Data Cleaning**         | \          | 50s         | 2mins 20s     | 13s           | 23s            | 3mins 46s          |
> | **In-depth EDA**          | \          | 41s         | 3mins 10s     | 12s           | 1min 10s       | 4mins 13s          |
> | **Feature Engineering**   | \          | 1min 13s    | 24s           | 15s           | 31s            | 2mins 23s          |
> | **Model Building, Validation, and Prediction** | \ | 57s         | 10mins 11s    | 35s           | 32s            | 12mins 15s         |
> | **Total**                 | 21s        | 4mins 28s   | 17mins 6s     | 1min 33s      | 3mins 41s      | **27mins 9s**      |
>
> From the detailed time cost, it is evident that the **Developer agent consumes the majority of time, accounting for 63% of the total**. This proportion becomes even higher for datasets with larger data volumes. The main time cost lies in running code (e.g., data processing, model training), while the other four agents show relatively consistent time consumption across different tasks.
>
> Additionally, due to AutoKaggle's stage-based execution and its comprehensive reporting feature, even for simple datasets, it performs detailed analyses, resulting in a baseline time cost. This is further influenced by the local environment used for these experiments. The speed of running code (e.g., data processing, model training) was limited by the suboptimal performance of the local hardware, thereby increasing AutoKaggle's overall runtime.
>
> To summarize, the primary reasons for the time cost are as follows:
>
> 1. **Comprehensive stage-based analysis**: AutoKaggle performs detailed analyses for each stage and generates thorough reports. Even for simple datasets, this results in a baseline time cost.
> 2. **Hardware performance limitations**: These experiments were conducted on a local machine, where the speed of running code was constrained by the hardware's suboptimal performance. If executed on a high-performance server, AutoKaggle's runtime would be significantly reduced.
>
> We assure you that in future experiments, we will verify performance on high-end servers.
>
> ---
>
> > **W10:** ML code generation should not be particularly challenging for GPT-4-mini, especially for Tasks 1-3. What are the main reasons for the failures in these cases since the result drops significantly?
>
> Thanks for your insightful question!
>
> We analyze the reasons behind GPT-4-mini's failures, and we find that almost all errors occurred due to its inability to correctly read the `train.csv` and `test.csv` files. This highlights a common issue faced by automated frameworks in real-world scenarios—failure to correctly locate file paths. If GPT-4-mini were directly provided with pre-loaded datasets, such as `train_df = pd.read_csv('train.csv')`, it has the capability to generate code to perform data cleaning. However, in real execution environments, its limited ability to comprehend long contexts prevents it from accurately identifying information about file locations within the given context, leading to errors at the very first step.
>
> This also reveals areas where AutoKaggle can be improved. We commit to further optimizing the design of AutoKaggle’s architecture to enable even relatively weaker base models to complete the entire data science pipeline in real-world environments.
>
> Thanks again for your valuable questions and comments! And I sincerely hope our explanation provides you with greater clarity.

---

> ### Author Response · Authors · 2024-12-02
>
> Dear Reviewer zZNf,
>
> Thank you for your time and effort in reviewing our paper, as well as for your constructive feedback and valuable questions. We sincerely appreciate the thoughtfulness you have brought to the review process.
>
> As the rebuttal period concludes today, we kindly ask if our responses meet your expectations or if further clarifications are needed. If they do address your concerns, we would greatly appreciate your consideration in reevaluating the score. Otherwise, we are happy to provide any additional clarifications within the remaining time.
>
> Thank you again for your valuable input, which has greatly contributed to improving our work.
>
> Best regards,
>
> Authors

---

> > ### Comment · Reviewer_zZNf · 2024-12-02
> >
> > Thank you for the additional results. After carefully reviewing the updates, I have decided to maintain my score.

---

> > > ### Author Response · Authors · 2024-12-02
> > >
> > > Dear Reviewer zZNf,
> > >
> > > Thank you for taking the time to carefully review the updates and for providing your thoughtful response. I appreciate your detailed consideration and respect your decision to maintain the score.
> > >
> > > If there are any additional points or clarifications you would like me to address in the future, I would be happy to provide further information.
> > >
> > > Thank you once again for your valuable feedback and for the effort you have put into reviewing my work.
> > >
> > > Best regards,
> > >
> > > Authors

---

### Official Review · Reviewer_1dns · 2024-11-04

**Soundness:** 3
**Presentation:** 3
**Contribution:** 2
**Rating:** 5
**Confidence:** 3

**Summary:**

This paper introduces AutoKaggle, a pipeline to automatically solve Kaggle Competitions. The authors use 5 subparts in a row: a reader, a planner, a developer, a reviewer, and a summarizer. They use LLMs with RAG to develop code-based solutions, with code running, units tests. They evaluate their method on 5 Kaggle competition benchmarks.

**Strengths:**

**Interesting problem.** With the LLMs (+RAG) becoming mature, the open source study of their integration into broader tools that can directly be applied to data science tasks, is the natural next step.

**Overall good presentation.** Even if some details are lacking to grasp the authors' exact contribution (notably in the figures), the overall presentation clearly demonstrates the problem and the approach set up to tackle it.

**Interesting metrics and ablation studies.**

**Weaknesses:**

**Lacking evaluation.** The evaluation is lacking comparison to existing AutoML baselines (*e.g. [1]) or explanations on why the authors are not comparing their method to any existing solution. If running such comparison is not possible at all, then the authors should provide explanations on why this is not feasible.
While detailed reports are provided on their methods and the different components, as this works apply existing techniques, its evaluation is its core contribution.
The authors should report (at least) the standard deviation, but e.g. violin plots to compare AutoKaggle's results of other kaggle competitors could help clearly situate where this automatic pipeline stands.

**Evaluation on a (previously) unknown dataset.** It seems that AutoKaggle has been designed to solve these datasets, so one cannot evaluate how much this method would transfer to another, previously unknown dataset.
It would be nice to provide the reader with how much out of the box your method is, maybe with a user study. It seems like its your core contribution, so having independent people trying AutoKaggle and commenting on how easy the setup and interaction is on a left out dataset would help people looking for such solutions.

**Figure 2 could be improved.** The figure could be split to separate the overall pipeline from details on some of its components. Most importantly, what part is using an LLM, what part is using a human expert ? This figure represents 70% of what the reader is looking for, it should provide first the overall intuition, and then enough details on specific core components that you want to highlight.

**You related work section is actually a background section.**
Your current related work covers some domains that are integrated within AutoKaggle. It thus feels more like a related work of your background section (what AutoKaggle builds upon). Is there any *e.g.* AutoML method that you can compare to ? Any method that addresses the same issue ?


[1] https://github.com/automl/CAAFE

**Questions:**

* Did you do any finetuning over the used models, notably LLMs or are you using frozen models ?
* Why cannot you compare to any existing baselines ?
* Have you optimized the creation of your pipeline using these 5 kaggle competitions, or have you left out some of them, to evaluate on competitions you did not know at design time ?

---

> ### Author Response · Authors · 2024-11-24
> **Thanks for your valuable comments! Authors' feedback [1/2]**
>
> Thanks for your valuable feedback and constructive comments. Below, we present our point-by-point response to the weaknesses and comments identified in our submission:
>
> > **W1:** **Lacking evaluation.** The evaluation is lacking comparison to existing AutoML baselines (*e.g. [1]) or explanations on why the authors are not comparing their method to any existing solution. If running such comparison is not possible at all, then the authors should provide explanations on why this is not feasible. While detailed reports are provided on their methods and the different components, as this works apply existing techniques, its evaluation is its core contribution. The authors should report (at least) the standard deviation, but e.g. violin plots to compare AutoKaggle's results of other kaggle competitors could help clearly situate where this automatic pipeline stands.
> >
> > **Q2:** Why cannot you compare to any existing baselines ?
>
> Thanks for your valuable feedback. For this issue, please refer to the first point in the **General Response - Common Problems**.
>
> ----
>
> > **W2：Evaluation on a (previously) unknown dataset.** It seems that AutoKaggle has been designed to solve these datasets, so one cannot evaluate how much this method would transfer to another, previously unknown dataset. It would be nice to provide the reader with how much out of the box your method is, maybe with a user study. It seems like its your core contribution, so having independent people trying AutoKaggle and commenting on how easy the setup and interaction is on a left out dataset would help people looking for such solutions.
> >
> > **Q3:** Have you optimized the creation of your pipeline using these 5 kaggle competitions, or have you left out some of them, to evaluate on competitions you did not know at design time ?
>
> Thanks for your valuable comments. We focus on the evaluation of AutoKaggle on unknown datasets. Note that AutoKaggle is not optimized for any specific datasets. AutoKaggle was developed based on the three toy datasets [1], not the Kaggle competitions we used for testing. We aim to provide a standard end-to-end processing solution for all tabular datasets. Another goal is to revise and optimize data science workflows so that data scientists can handle their daily tasks more efficiently.
>
> We chose Kaggle competitions as our testing platform during the evaluation process because they are closely related to real-world data science applications. In Section 3.1-Task Selection, we explained how we selected eight evaluation datasets. These datasets cover a variety of data science tasks, including classification and regression with single-target and multi-target variables, to ensure a comprehensive assessment of AutoKaggle's capabilities.
>
> In addition, our approach demonstrates robust performance and a high valid submission rate across various tasks. Our submission success rate (83%) has significantly improved across all 8 Kaggle tasks compared to the AIDE (28%). This result validates AutoKaggle's generalization capability and effectiveness.
>
> Regarding the usability of AutoKaggle, we plan to conduct a user study where independent users will try AutoKaggle in their daily data science scenarios. They will provide feedback on the ease of setup and interaction. This user study will help researchers better understand the applicability of our approach in different contexts. However, this is an independent research topic, with quantitative and qualitative evaluation methods fundamentally different from autonomous agents and multi-agent systems. We commit to report this aspect systematically in our future studies in information system or human-computer interaction conferences. Thanks again for the suggestions, and we will continue striving to improve our research.
>
> [1] https://github.com/geekan/MetaGPT/tree/2b160f294936f5b6c29cde63b8e4aa65e9a2ef9f/examples/di

---

> > ### Comment · Reviewer_1dns · 2024-11-25
> > **Thank you for your clarifications and extra experiments**
> >
> > Hi. Thank you for your additional work.
> > Unfortunately, this late answer does not allow me to check everything in details.
> > My main question/concern is about the comparison with the new AIDE baseline. You report scores on each task, but it somehow gets a score of 0 on task 8. Why is that ?
> > I think that because of this one outlier, the reported mean is biased. What about the median?
> > If I'd consider task 8 to be an outlier, what would be the conclusion of your experimental evidence ?
> >
> > I do not think that the paper should get rejected because it might perform worse than another approach, there might be qualities of AutoKaggle that AIDE does not possess, that would justify its selection on such tasks.
> >
> > I think that the user study would be an amazing addition to this work. I will augment my score before the end of the rebuttal, but I want to reevaluate the paper to do so, for which, I don't have the time now.
> >
> > Could you again list a short summary of the modifications (due to any reviewer's concern) ?
> > Another great improvement for next time is if you write your modifications in e.g. blue in the paper, such that the reviewers can spot them easily (for your next rebuttal).

---

> ### Author Response · Authors · 2024-11-24
> **Thanks for your valuable comments! Authors' feedback [2/2]**
>
> > **W3:** **Figure 2 could be improved.** The figure could be split to separate the overall pipeline from details on some of its components. Most importantly, what part is using an LLM, what part is using a human expert ? This figure represents 70% of what the reader is looking for, it should provide first the overall intuition, and then enough details on specific core components that you want to highlight.
>
> Thanks for your valuable feedback. We apologize for the confusion in our writing and figures. We have revised the main text and adjusted Figure 1 accordingly.
>
> 1. Throughout the process described in the main text, AutoKaggle operates without any human involvement. In our evaluation, we assessed only the performance of autonomous multi-agents, ensuring no human intervention to maintain the fairness and objectivity of our assessment.
> 2. In Appendix D.5, we have additionally designed a Human-in-the-loop module for the model. As we replied before, we will conduct a user study where independent users will try AutoKaggle in their daily data science scenarios. We designed this Human-in-the-loop module to support our future research endeavors.
>
> ---
>
> > **W4:** **You related work section is actually a background section.** Your current related work covers some domains that are integrated within AutoKaggle. It thus feels more like a related work of your background section (what AutoKaggle builds upon). Is there any *e.g.* AutoML method that you can compare to ? Any method that addresses the same issue ?
>
> Thanks for your good questions. We have added a discussion of existing work in the Section 4-Related Work (Lines 508-514).
>
> ---
>
> > **Q1:** Did you do any finetuning over the used models, notably LLMs or are you using frozen models ?
>
> Thanks for your good question. Without any fine-tuning, the agents in AutoKaggle are created directly using OpenAI's official APIs. Section 3.1, Experiment Details, explains the models underpinning the different agents in AutoKaggle. Specifically, the Reader, Reviewer, and Summarizer are based on the GPT-4o-mini model, the Developer is based on the GPT-4o model, and the Summarizer utilizes the GPT-4o/o1-mini model.
>
> Thanks again for your suggestions and valuable feedback, and I hope our explanation provides you with greater clarity.

---

> ### Author Response · Authors · 2024-11-25
>
> Dear Reviewer,
>
> Thank you very much for taking the time and effort to review our manuscript and providing such valuable feedback. As the discussion phase between authors and reviewers is nearing its conclusion, we would like to confirm whether our responses have adequately addressed your concerns.
>
> We provided detailed answers to each of your comments one day ago, and we sincerely hope that our responses have sufficiently clarified your concerns. If you have any remaining doubts or require further clarification, please do not hesitate to let us know. We are more than willing to continue the discussion to address any questions you may have.
>
> Thank you once again for your time and assistance!
>
> Best Regards,
>
> Authors

---

> ### Author Response · Authors · 2024-11-29
> **Thanks for your valuable comments! Authors' feedback [1/2]**
>
> Thanks for your valuable suggestions and guidance! Below I will respond to each of your questions one by one:
>
> > **Q1:**  You report scores on each task, but it somehow gets a score of 0 on task 8. Why is that ? I think that because of this one outlier, the reported mean is biased. What about the median? If I'd consider task 8 to be an outlier, what would be the conclusion of your experimental evidence ?
>
> Thanks for your question! Task 8 indeed has unique characteristics as it is a multi-target regression problem, while the first seven tasks are all single-target regression problems. For each task, we conducted five repeated runs under identical settings. If all runs failed, it indicates that the framework faces inherent difficulties in handling the task, rather than isolated anomalies. Compared to single-target regression, multi-target regression requires more complex feature interactions and processing. However, AIDE typically employs uniform encoding and transformation without detailed planning for feature interactions, which is the primary reason for its poor performance on Task 8.
>
> Since multi-target problems are relatively rare in Kaggle datasets, we reviewed 50 competition datasets and found only one such task. We performed additional evaluations on this dataset (Task 9), with the results shown below:
>
> | **Task** | **Framework** | **Valid Submission** | **Comprehensive Score** |
> |----------|---------------|-----------------------|--------------------------|
> | Task 9   | AutoKaggle    | 0.80                 | 0.752                    |
> |          | AIDE          | 0.20                 | 0.452                    |
>
> As shown, AIDE still underperformed, further demonstrating its challenges in handling rare multi-target regression problems. Moreover, since AIDE does not provide detailed logic for its solution generation (as it uses a one-stop generation approach), we could not analyze its behavior patterns further.
>
> It is worth noting that even if Task 8 is considered an outlier and excluded, AutoKaggle still outperforms AIDE by 0.17 on the Valid Submission metric and by 0.08 on the Comprehensive Score metric, which reinforces its superior performance compared to AIDE.
>
> [1] https://www.kaggle.com/competitions/playground-series-s3e26
>
> ---
>
> > **Q2:** I think that the user study would be an amazing addition to this work.
>
> Thanks for your valuable suggestion! Based on your advice, we evaluated the solutions of AIDE and AutoKaggle across eight criteria and invited five graduate students with Kaggle experience to assess them from a user perspective. The evaluation results are as follows:
>
> | **Criteria/Framework**  | **AIDE** | **AutoKaggle** | **Winner**       |
> |--------------------------|----------|----------------|------------------|
> | **Code Length**          | 5        | 2.8            | AIDE             |
> | **Modularity**           | 2        | 4.4            | AutoKaggle       |
> | **External Dependencies**| 4.8      | 3.6            | AIDE             |
> | **Feature Richness**     | 2.4      | 5              | AutoKaggle       |
> | **Comment Coverage**     | 3.2      | 4.4            | AutoKaggle       |
> | **Code Reusability**     | 2.4      | 3.8            | AutoKaggle       |
> | **Comprehensibility**    | 3.8      | 4.2            | AutoKaggle       |
> | **Overall (Average)**    | 3.2     | 4.0           | AutoKaggle       |
>
> We have provided part of the solutions from AIDE and AutoKaggle for the same problem in [1], and the detailed user evaluation in [2] for your reference.
>
> From the results, AIDE performed better in **Code Length** and **External Dependencies**, while AutoKaggle excelled in **Modularity**, **Feature Richness**, **Comment Coverage**, **Code Reusability**, **Comprehensibility**, and the overall score.
>
> In user evaluations, AIDE’s solutions were praised for their **conciseness, fewer external dependencies, and ease of understanding**, but were criticized for their **lack of modularity, limited functionality, fewer comments, and poor code reusability**. On the other hand, AutoKaggle’s solutions were noted to be **lengthy and reliant on custom tools**, but they stood out for their **clear modular design, rich functionality, ability to generate data analysis visualizations, detailed step-by-step comments, and strong code reusability**, which resulted in higher overall user satisfaction.
>
> [1] https://anonymous.4open.science/r/AutoKaggle-B8D2/user_study/comparison_results
>
> [2] https://anonymous.4open.science/r/AutoKaggle-B8D2/user_study/README.md

---

> ### Author Response · Authors · 2024-11-29
> **Thanks for your valuable comments! Authors' feedback [2/2]**
>
> > **Q3:** Could you again list a short summary of the modifications (due to any reviewer's concern) ? Another great improvement for next time is if you write your modifications in e.g. blue in the paper, such that the reviewers can spot them easily (for your next rebuttal).
>
> Thanks for your valuable suggestions! Here is a summary of the modifications we made:
>
> 1. **Added comparison experiments with the AIDE framework**: We included results and corresponding analyses of the AutoKaggle framework based on o1-mini in Section 3.2 (Main Results).
> 2. **Revised evaluation metrics**: Metrics were updated to Made/Valid Submission (referencing MLE-Bench) and Comprehensive Score (referencing [3]).
> 3. **Added Appendix B: Error Analysis**: This section provides a detailed analysis of the error distribution encountered during AutoKaggle's execution of data science tasks and describes its error correction methods.
> 4. **Added Appendix E: Case Study**: Using the Titanic competition on Kaggle as an example, we detailed the staged workflow of AutoKaggle along with some intermediate results to help clarify its technical details.
> 5. **Enhanced the README file in the anonymous GitHub repository**: We improved the explanation of how to use AutoKaggle and provided example results in `multi_agents/example_results/` for review.
> 6. **Added a user study part**: Five graduate students with computer science backgrounds and Kaggle experience evaluated the solutions generated by AutoKaggle and AIDE across seven dimensions. The detailed results can be found in the `user_study/README.md` file in the anonymous GitHub repository.
>
> Additionally, we have marked all modified sections in the paper with red for clarity. Thanks again for your valuable suggestions and kind guidance! And I sincerely hope our explanation provides you with greater clarity.

---

> ### Author Response · Authors · 2024-12-02
>
> Dear Reviewer 1dns,
>
> Thank you for your time and effort in reviewing our paper, as well as for your constructive feedback and valuable suggestions. We sincerely appreciate the thoughtfulness you have brought to the review process.
>
> As the rebuttal period concludes today, we would like to kindly remind you of your earlier comment: "I think that the user study would be an amazing addition to this work. I will augment my score before the end of the rebuttal, but I want to reevaluate the paper to do so, for which, I don't have the time now." We truly hope that our recent response has addressed your concerns and clarified the points raised.
>
> If our clarifications meet your expectations, we would greatly appreciate your consideration in reevaluating the score. However, if additional questions remain, we would be happy to provide further clarifications within the limited time left.
>
> Thank you again for your valuable input, which has greatly contributed to improving our work.
>
> Best regards,
>
> Authors

---

### Author Response · Authors · 2024-11-24
**General Response**

We extend our sincere appreciation to each of the three reviewers for the time dedicated to reviewing our manuscript and for the constructive feedback provided. Your insights and critiques are invaluable.

**Common Problems**

In reviewing the feedback from the reviewers, we identified and addressed two recurring themes:

1. Lack of Evaluation and Baseline Comparison for AutoKaggle

We have added comparative experiments with the AIDE framework and included results and corresponding analysis of the AutoKaggle framework based on o1-mini in Section 3.2-Main Results. AIDE[1] is the top-performing framework in MLE-Bench[2]. The new main experimental results are presented in Section 3.2-Main Results(Table 1). The results demonstrate that AutoKaggle significantly outperforms the AIDE framework in terms of valid submission rate and comprehensive score across the eight datasets we evaluated.

1. Special Optimization of AutoKaggle for Evaluation Datasets

AutoKaggle is not specifically optimized for any particular dataset. It is designed to provide a general end-to-end solution for all types of tabular datasets. Our goal is to simplify and optimize the data science workflow, enabling data scientists to handle daily tasks more efficiently. The development of AutoKaggle was based on three toy datasets from [3], rather than the Kaggle competitions we used for testing.

**Other Revisions**

In addition to incorporating a comparison with AIDE, we have made the following revisions to the article based on the reviewers' feedback:

1. Revised Evaluation Metrics: We updated the metrics to include Made/Valid Submission (sourced from MLE-Bench) and Comprehensive Score (sourced from [4]). The calculation method for the Comprehensive Score has been adjusted to 0.5 × Valid Submission + 0.5 × ANPS to better align with the evaluation of our framework.
2. Added Appendix B-Error Analysis: This section provides a detailed analysis of the error distribution encountered by AutoKaggle during the completion of data science tasks. It also describes the code correction methods used within AutoKaggle.
3. Added Appendix E-Case Study: We included a case study using the Titanic competition from Kaggle. This section details the phased workflow of AutoKaggle and presents some intermediate results to enhance understanding of the technical details of AutoKaggle.
4. Enhanced README in Anonymous GitHub Repository: We improved the README file to better explain how to use AutoKaggle. Additionally, sample results have been provided in the multi_agents/example_results/ directory for review.

More questions have been answered in the point-to-point responses. We hereby assure you that all of them will be resolved in the manuscript.

Collectively, we anticipate that, within our point-to-point responses, the detailed explications of our method and the inclusion of supplementary baseline comparisons will clarify the misunderstandings and substantively address the feedback.

We extend our heartfelt gratitude once again to all the reviewers for their meticulous and insightful critiques.

[1] AIDE: https://github.com/WecoAI/aideml

[2] Chan J S, Chowdhury N, Jaffe O, et al. Mle-bench: Evaluating machine learning agents on machine learning engineering[J]. arXiv preprint arXiv:2410.07095, 2024.

[3] Hong S, Lin Y, Liu B, et al. Data interpreter: An LLM agent for data science[J]. arXiv preprint arXiv:2402.18679, 2024.

[4] https://github.com/geekan/MetaGPT/tree/2b160f294936f5b6c29cde63b8e4aa65e9a2ef9f/examples/di

---

### Meta-Review · Area_Chair_BU2a · 2024-12-20

**Metareview:**

The paper presents AutoKaggle, an LLM creating a multi-agent system together with a library of hand-crafted ML tools in order to solve Kaggle problems. In my opinion, the reviewers did a very thorough job and have presented salient arguments about the suitability of this paper for IJCAI in its current form. While it is interesting to see that an agentic LLM can help with Kaggle competitions, one reviewer points out that a comparison to existing AutoML approaches is missing. Indeed, the authors added a comparison to AIDE. However, AutoML is not only "an LLM agent that generates solutions for machine learning tasks" (as written on the AIDE github page) but also portfolio and other approaches to automatize (parts of) ML. So, a larger discussion (and comparison) is in place. As it turn out Kaggle is partnering with the International Conference on Automated ML,see https://www.kaggle.com/automl-grand-prix. Another reviewer points out that some design choices and arguments for motivation are missing; currently, it reads more like what the authors have done, but it is not well placed into the research landscape. More importantly, an evaluation across different types of modalities (time series, text classification, image object detection, ...) is missing or argued why this is currently not so important. One reviewer also pointed out that an ablation study is missing, showing that the hand-crafted tools are not doing the job in the end. So, while the direction is super interesting, it is too early for publication, but we would like to encourage the authors to push  for one of the next venues. Please note that the overall judgment should not be taken as a statement regarding the usefulness of your research.

**Additional Comments On Reviewer Discussion:**

The discussion arose from issues raised in the reviews. Issues touched were (missing) baselines, user study, ablation study, and human-written unit-test. Overall, the rebuttal / discussion did not change the mind of the reviewers.

---

### Decision · Program_Chairs · 2025-01-22

Reject